# Novel Gene Signatures Predictive of Patient Recurrence-Free Survival and Castration Resistance in Prostate Cancer

**DOI:** 10.3390/cancers13040917

**Published:** 2021-02-22

**Authors:** Jun A, Baotong Zhang, Zhiqian Zhang, Hailiang Hu, Jin-Tang Dong

**Affiliations:** 1Department of Genetics and Cell Biology, College of Life Sciences, Nankai University, 94 Weijin Road, Tianjin 300071, China; 1120170387@mail.nankai.edu.cn; 2Department of Human Cell Biology and Genetics, School of Medicine, Southern University of Science and Technology, 1088 Xueyuan Road, Shenzhen 518055, China; zhangzq@sustc.edu.cn; 3Emory Winship Cancer Institute, Department of Hematology and Medical Oncology, Emory University School of Medicine, 1365-C Clifton Road, Atlanta, GA 30322, USA; baotong.zhang@emory.edu; 4Department of Biochemistry, School of Medicine, Southern University of Science and Technology, 1088 Xueyuan Road, Shenzhen 518055, China; huhl@sustech.edu.cn

**Keywords:** prostate cancer, castration-resistant prostate cancer (CRPC), prognosis, gene signature, recurrence-free survival

## Abstract

**Simple Summary:**

Molecular signatures predictive of recurrence-free survival (RFS) and castration resistance are critical for treatment decision-making in prostate cancer (PCa), but the robustness of current signatures is limited. This study aims to identify castration-resistant PCa (CRPC)-associated genes and develop robust RFS and CRPC signatures. Among 287 genes differentially expressed between localized CRPC and hormone-sensitive PCa (HSPC) samples, 6 genes constituted a signature (CRPC-derived prognosis signature, CRPCPS) that predicted RFS. Moreover, a 3-gene panel derived from the 6 CRPCPS genes was capable of distinguishing CRPC from HSPC. The CRPCPS predicted RFS in 5/9 cohorts in the multivariate analysis and maintained prognostic in patients stratified by tumor stage, Gleason score, and lymph node metastasis status. It also predicted overall survival and metastasis-free survival. Notably, the signature was validated in another six independent cohorts. These findings suggest that these two signatures could be robust tools for predicting RFS and CRPC in clinical practice.

**Abstract:**

Molecular signatures predictive of recurrence-free survival (RFS) and castration resistance are critical for treatment decision-making in prostate cancer (PCa), but the robustness of current signatures is limited. Here, we applied the Robust Rank Aggregation (RRA) method to PCa transcriptome profiles and identified 287 genes differentially expressed between localized castration-resistant PCa (CRPC) and hormone-sensitive PCa (HSPC). Least absolute shrinkage and selection operator (LASSO) and stepwise Cox regression analyses of the 287 genes developed a 6-gene signature predictive of RFS in PCa. This signature included *NPEPL1*, *VWF*, *LMO7*, *ALDH2*, *NUAK1*, and *TPT1*, and was named CRPC-derived prognosis signature (CRPCPS). Interestingly, three of these 6 genes constituted another signature capable of distinguishing CRPC from HSPC. The CRPCPS predicted RFS in 5/9 cohorts in the multivariate analysis and remained valid in patients stratified by tumor stage, Gleason score, and lymph node status. The signature also predicted overall survival and metastasis-free survival. The signature’s robustness was demonstrated by the C-index (0.55–0.74) and the calibration plot in all nine cohorts and the 3-, 5-, and 8-year area under the receiver operating characteristic curve (0.67–0.77) in three cohorts. The nomogram analyses demonstrated CRPCPS’ clinical applicability. The CRPCPS thus appears useful for RFS prediction in PCa.

## 1. Introduction

Prostate cancer (PCa) is one of the most commonly diagnosed malignancies in men worldwide [1]. For example, in 2019 alone, an estimated 174,650 new cases of PCa were diagnosed, and 31,620 men died of PCa in the United States [2]. While most prostate cancers are benign, some progress to relapse or metastasize to other organs after initial surgery and radiation therapies, presenting a series of aggressive disease characteristics and killing patients. It is estimated that 17–33% of PCa patients who have received radical prostatectomy will experience a biochemical recurrence, and approximately 30% of individuals develop metastatic disease [3,4,5,6,7,8]. Androgen deprivation therapy (ADT), a mainstay of treatment against locally advanced or metastatic PCa, suppresses the growth of PCa via a decline in the circulating testosterone or functional inhibition of androgen receptor (AR) [9,10]. However, most ADT-treated patients eventually develop a castration-resistant PCa (CRPC) after 12–24 months of ADT treatment, and the median survival time of CRPC patients is approximately 14 months [11].

Previous studies have demonstrated that some clinicopathological characteristics can provide important prognostic information for monitoring disease progression in PCa, including PSA (prostate-specific antigen) level, Gleason score, pathological tumor stage, and lymph node status. However, the prediction power of these conventional markers is often limited, especially for patients with an ambiguous clinical diagnosis or intermediate grades or stages. With the availability of cancer genome sequences at both DNA and RNA levels, numerous molecular indicators have been developed to predict disease progression, including recurrence-free survival (RFS). Some of the biomarkers are based on a single gene’s alterations, such as those of *BRCA2* [12,13], *miR-301a* [14], *AXIN2* [15], and *NRP1* [16]. Some others are multi-gene-based, including the Oncotype DX (17 genes) [17], the Prolaris (31 genes) [18], and the miRNA-based prognostic ratio MiCaP model (4 miRNAs) [19]. Such molecular biomarkers have been shown to enhance the accuracy of PCa aggressiveness prediction. 

However, several obvious limitations for the application of such molecular biomarkers are noticed. Firstly, most of the gene signatures were developed from a defined set of genes related to specific cellular hallmarks of cancer, including those of autophagy [20], hypoxia [21], matrix metalloproteinases [22], immune infiltration [23], and cell cycle [24]. Secondly, only a fraction of those signatures have been assessed for their application in predicting RFS in PCa patients with a specific clinicopathological parameter, including tumors with the Gleason score of seven and those with or without lymph node metastasis [25,26]. PCa is highly heterogeneous, so many complicated molecular mechanisms are responsible for disease relapse even in tumors with the same tumor stage, grade, or lymph node status. Thirdly, some of the signatures contain many genes (e.g., 100-gene set [27]), which could be a barrier in their clinical application. In addition, a limited number of patients have been included in most previous studies, which may compromise the prediction power or reliability. 

Regarding the signatures for CRPC, only a small number of studies are available and even fewer compared genome-wide expression profiles between hormone-sensitive prostate cancer (HSPC) and CRPC. Furthermore, although many studies have developed molecular signatures that can predict disease recurrence, none of them have been used to distinguish CRPC from HSPC tumors.

In this study, we attempted to develop robust signatures for RFS and CRPC prediction in PCa. We identified 287 differentially expressed genes (DEGs) between localized CRPC and HSPC tumors from 4 GEO (Gene Expression Omnibus) datasets. Based on these DEGs, we developed a 6-gene CRPC prognostic signature (CRPCPS) that can predict RFS. Moreover, a 3-gene panel derived from the 6-gene CRPCPS distinguished CRPC from HSPC with high specificity and sensitivity in both the training and validation cohorts. This CRPCPS was further validated in another 6 independent datasets from different platforms. Notably, the CRPCPS remained valid across different clinicopathological subgroups, including tumors with the same Gleason scores, pathological tumor stage, and lymph node status. Its robustness and applicability in clinical practice were demonstrated by the AUC (area under the receiver operating characteristic curve) and nomogram analyses, respectively. The robust performance of these signatures on predicting RFS and CRPC indicates their universal applicability in clinical practice.

## 2. Materials and Methods

### 2.1. Patient Cohorts

Four gene expression profiles of PCa, including GSE6811 [28], GSE2443 [29], GSE28680 [30], and GSE70768 [27], were downloaded from the Gene Expression Omnibus (GEO) (http://www.ncbi.nlm.nih.gov/geo accessed on 5 September 2019) and used to identify DEGs between localized CRPC and HSPC samples (Appendix A). The criteria for selecting these datasets included: (1) Gene expression data must be available for both CRPCs and HSPC tumors, (2) both CRPCs and HSPC tumors must be derived from the primary site (metastases and those with ambiguous sources were excluded), and (3) at least 5000 genes must be included when the microarray platform is used for expression profiling. In general, a CRPC is defined by the consecutive elevation of serum PSA levels and/or the appearance of progressive measurable diseases such as enlargement of a primary tumor or the detection of metastasis. Of the 5 datasets (including the GSE35988 dataset below) with CRPC tumors, only two provided CRPC definition (GSE2443 and GSE6811), and both datasets used one or more of the above criteria. In the other three datasets, a patient’s CRPC status was provided without specifying how it was defined. CRPC and HSPC statuses of all patients were retrieved from the series matrix files of all datasets.

For model development, RNA sequencing (RNA-seq) data of 499 PCa patients and corresponding clinical information in the Cancer Genome Atlas (TCGA) were obtained from the UCSC (University of California, Santa Cruz) Xena public data hub (https://xenabrowser.net/ accessed on 22 July 2018). Of these 499 samples, 63 did not have recurrence information and thus were excluded for analysis. The remaining 436 PCa samples were randomly divided into the training set (*n* = 305) and the internal validation set (*n* = 131). An additional 6 cohorts were enrolled as the external validation sets, including those of the Belfast (GSE116918, *n* = 248) [31], MSKCC (The Memorial Sloan Kettering Cancer Center, *n* = 140) [32], CPC-GENE (Canadian Prostate Cancer Genome Network, GSE84042, *n* = 73) [33], Cambridge (GSE70768, *n* = 106) [27], Stockholm (GSE70769, *n* = 92) [27], and GSE54460 (*n* = 91) [34]. All expression profile data were from either the GEO database or the cBio Cancer Genomics Portal (https://www.cbioportal.org/ accessed on 11 October 2019). The clinicopathological information for the enrolled cohorts is summarized in Table 1.

For CRPC diagnosis analysis, we used the GSE70768 dataset [27] from the DEG analysis above, which included more tumor samples than other datasets. We also included the GSE70769 dataset [27], which had a large number of HSPC tumors and used the same expression profiling platform as the GSE70768 dataset. For validation, dataset GSE35988 [35] was used, and metastatic CRPC tumors in the validation dataset (GSE35988) were also included due to primary CRPC tumors’ rarity. The design and workflow of this study are shown in Figure 1.

### 2.2. Gene Expression Data Preprocessing

The series matrix files and their platform annotation information were downloaded, and probes with missing values for more than half of the samples or detected with more than one gene were excluded for analysis. Due to the variation in gene symbols across different microarray platforms, probes were assigned to their Entrez identification numbers for analysis according to each platform’s annotated files. If multiple probe sets corresponded to the same Entrez identification number, they were integrated using the arithmetic mean to account for a gene’s expression level. Quantile normalization was then performed for mining DEGs. For the TCGA dataset, genes with an expression level of 0 (not detected) in >10% of patients were removed.

### 2.3. Identification of CRPC-Associated Genes

The R package “limma” (version 3.44.1) [36] was applied to identify DEGs between localized CRPC and HSPC tumors. A *p*-value smaller than 0.05 was used as the cut-off point for initial DEG identification. Due to the high variability among different microarray platforms, gene integration analysis with the R package “RobustRankAggreg” (Robust Rank Aggregation (RRA), version 1.1) [37] was used to select more robust DEGs from the initial DEGs obtained in the previous step. Genes with |log2 (fold change)| > 0.5 and *p* < 0.05 in the RRA analysis were considered as the CRPC-associated genes. Subsequently, the R package “RCircos” (version 1.2.1) [38] was used to visualize the expression patterns of different microarrays and chromosomal locations for the top 100 DEGs sorted by their *p*-values. Gene Ontology (GO) enrichment and Kyoto Encyclopedia of Genes and Genomes (KEGG) pathway analyses were performed by the R package “clusterProfiler” (version 3.14.3) [39]. GO terms or KEGG pathways with *p* < 0.05 were considered statistically significant and visualized by “dotplot”.

### 2.4. Construction and Validation of a CRPC-Derived Prognostic Signature (CRPCPS)

The 436 PCa patients in the TCGA dataset were randomly assigned to training (*n* = 305) and internal validation cohort (*n* = 131) at a 7:3 ratio. Univariate Cox regression analysis was performed to assess genes associated with RFS in the TCGA training cohort. The CRPC-associated genes significantly associated with RFS (*p* < 0.05) were considered the candidate gene set. Subsequently, the least absolute shrinkage and selection operator (LASSO) method [40] by glmnet (version 4.0.2) R package for variable selection in a Cox regression model was used to reduce the dimensionality, and a forward and backward variable selection procedure [41] via the ‘stepAIC’ function from the R package ‘MASS’ (version 7.3.53) was further conducted to select the optimal gene combination with the lowest Akaike information criteria (AIC) as a CRPC-derived prognostic signature (CRPCPS).

Subsequently, a patient’s risk score for RFS was built based on a linear combination of the regression coefficient derived from the multivariate Cox regression model and the expression level of the optimized genes (i.e., the signature genes). The risk score was computed as follow: Risk score = (gene1 expression × gene1 coefficient) + (gene2 expression × gene2 coefficient) + … + (gene N expression × gene N coefficient). PCa patients were classified into high-risk and low-risk groups by the best cut-off values determined by the function “surv_cutpoint” of the survminer (version 0.4.8) R package. The Kaplan–Meier (K-M) method and log-rank test were applied to assess differences between the high-risk and low-risk groups via the ‘survfit’ and ‘survdiff’ functions of the R package ‘survival’ (version 3.2.7) [42]. 

Model performance was evaluated by discrimination and calibration [43]. Discrimination was assessed using Harrell’s concordance index (C-index) adjusted through 1000 bootstrap resamples by the “validate” function of rms (version 6.0.1) R package. The time-dependent receiver operating characteristic (ROC) curve performed by survival-ROC (version 1.0.3) R package [44] was used to assess the efficiency of the CRPCPS. Bias-corrected calibration for 3-, 5-, and 8-year RFS probability using 1000 bootstrap resamples was further applied to visualize the consistency between actual and estimated RFS probabilities by the rms (version 6.0.1) R package (https://hbiostat.org/R/rms/ accessed on 12 October 2020). The predictive model was then validated in the internal TCGA validation cohort and another six independent cohorts.

The integrated area under the curves (IAUC) of the model (i.e., CRPCPS) was compared to another three published transcriptomic signatures [20,45,46] by survcomp (version 1.36.1) package. The prediction error was estimated by the pec (version 2020.11.17) R package (https://cran.r-project.org/web/packages/pec/ accessed on 21 November 2020).

### 2.5. Construction of a Diagnostic Model for CRPC

To investigate whether the CRPCPS could be applied for predicting CRPC occurrence, two datasets from the GEO database, GSE70768 and GSE70769, were pooled together, and the combined dataset was then adjusted for batch effect through the “ComBat” function of sva (version 3.34.0) R package [47] and assigned as the training set. Logistic regression [41] was applied to the training set to build a logistic regression model, with all CRPCPS genes as covariates. Due to a small percentage (6%) of CRPC samples in the training set, which leads to a learning bias to the majority class, SMOTE (synthetic minority over-sampling technique) was used to deal with the so-called class imbalance problems, as previously reported [48] by DMwR (version 0.4.1) R package. The forward stepwise selection was then applied to the model to generate the final diagnostic model. For each gene in the diagnostic model, its unbiased coefficient estimate (from the training set) was multiplied by the gene’s expression level. All genes’ multiplications were added to generate the CRPC-diagnosis score for a tumor. The predictability of the model was then evaluated by area under the curve (AUC) of ROC. Calibration was examined by using the Brier score [49] and the Spiegelhalter *z* test [50].

Another transcriptional profile from the GEO database, GSE35988, which included 59 HSPC and 35 CRPC samples, was used for the validation of the model.

### 2.6. Establishment and Assessment of the Nomogram

To further evaluate the usefulness of the CRPCPS, we used patients with detailed clinicopathological information in the TCGA dataset, including the age at diagnosis, Gleason score, pathological tumor stage, and lymph node status. Serum PSA level was excluded because few patients had a higher pre-operation PSA level (PSA > 10 ng/mL). All clinicopathological characteristics were evaluated as categorical variables. Univariate and multivariate Cox regression analyses were performed to a variant’s association with RFS. The same TCGA training cohort described above was used to establish a nomogram for predicting 3-, 5-, and 8-year RFS rates of PCa patients. The same TCGA validation cohort and the entire TCGA cohort were used for validation. ROC curve and calibration plot were constructed using rms (version 6.0.1) R package to evaluate the nomogram’s efficiency. 

### 2.7. Functional Enrichment Analysis

To explore the potential molecular events involving CRPCPS, the gene set variation analysis (GSVA) (version 1.34.0) R package [51] was applied to gene sets that were differentially expressed between the high- and low-risk groups and involved in biological processes and signaling pathways. Two gene sets in the MsigDB v7.1 database (http://www.broad.mit.edu/gsea/msigdb/ accessed on 28 May 2020), i.e., c5.bp.v7.1.symbols.gmt and c2.cp.kegg.v7.1.symbols.gmt, were downloaded and used as reference gene sets [52]. An adjusted *p*-value of 0.05 was set as the cut-off point for significant enrichment.

### 2.8. Statistical and Other Computational Analyses

All statistical analyses were conducted using the R software (Windows version 3.6.3; https://www.r-project.org/ accessed on 7 March 2020), and Bioconductor (version 3.10) [53]. The univariate and multivariate Cox proportional hazards regression analyses were performed using the ‘coxph’ function of the “survival” (version 3.2.7) R package [54,55]. The heatmaps, boxplots, forest plots, and calibration plots were drawn using the R software. Statistical differences between the two groups were assessed using the Wilcoxon test. All statistical tests were two-sided, and *p* < 0.05 was considered statistically significant.

## 3. Results

### 3.1. Integrated Screening for Robust CRPC-Associated Genes

Four GEO datasets were used for the identification of robust CRPC-associated genes (Appendix A). Using the “limma” R package, we normalized expression data from datasets GSE6811, GSE2443, GSE28680, and GSE70768, and identified 2588, 1331, 2221, and 10,786 DEGs between localized CRPC and HSPC tumors respectively, with a cut-off of *p* < 0.05 (Figure 2a). Integration of these DEGs by the RRA method resulted in 287 DEGs (|log2FC| ≥ 0.5, *p* < 0.05), 93 of which were upregulated and 194 downregulated in CRPCs (Appendix A). Of these 287 CRPC-associated genes, 35 were identified in one previous study [29], 47 were identified in another study [28], and the remaining 207 have not been identified in a genome-wide analysis (Appendix A), which is marked in bold in Appendix A. 

The top 20 upregulated and the top 20 downregulated genes in CRPC are shown as hierarchical cluster heatmaps (Figure 2b). For the top 100 DEGs between CRPC and HSPC, their expression patterns across the four datasets used for analysis, along with their chromosomal locations, are shown in a circos plot (Figure 2c). The location of these 100 DEGs involves all chromosomes except for the Y chromosome. The top 5 upregulated genes included *COL5A2*, *THBS2*, *CDC20*, *COL1A1*, and *COL3A1*, and they are located on chromosomes 2, 6, 1, 17, and 2, respectively. The top 5 downregulated genes (*MSMB*, *DUSP1*, *CCN1*, *MT1X*, and *TRPM8*) are located on chromosomes 10, 5, 1, 16, and 2, respectively.

GO enrichment and KEGG pathway analyses were performed to further elucidate the potential biological function and the promising signaling pathways involving the entire 287 DEGs. Several biological processes were highly enriched in the GO analysis, including the extracellular matrix organization, the epithelial cell proliferation, the response to steroid hormones, and the regulation of ERK1 (Extracellular signal-related kinase 1) and ERK2 cascades. In the KEGG pathway analysis, the DEGs participated in diverse hormonal signaling pathways, including parathyroid hormone’s synthesis, secretion, and action, cortisol synthesis and secretion, aldosterone synthesis and secretion, and some cancer-associated pathways such as the PI3K-Akt (Phosphatidylinositide 3-kinases-protein kinase B) signaling pathway, focal adhesion pathway, and several others that are involved in small cell lung carcinomas (Figure 2d).

### 3.2. Development of a CRPC-Derived Six-Gene Signature That Predicts RFS

As described in the Section 2, the univariate Cox regression analysis was performed to assess genes associated with RFS in the TCGA training cohort (Table 1), and 24 of the 287 CRPC-associated DEGs were identified for a significant association with RFS in PCa (*p* < 0.05) (Appendix A). These 24 genes were further analyzed using the LASSO Cox regression method in the training cohort, which minimized the potential overfitting problems and determined the minimum criteria. Nine of the twenty-four genes remained in the model at the optimal tuning parameter (λ) (log (lambda.min) = 0.02273839) (Figure 3a,b). For the nine genes, multivariate Cox regression analysis was used to fit them into an initial model. A forward and backward variable selection was then applied and resulted in the appropriate gene combination of 6 genes (*NPEPL1*, *VWF*, *LMO7*, *ALDH2*, *NUAK1*, and *TPT1*) (Table 2). Kaplan–Meier analysis demonstrated that patients with a higher expression level of *ALDH2*, *LMO7*, or *TPT1* had a better RFS than those with a lower level (Figure 3c, *p* < 0.05). For *NPEPL1*, *NUAK1*, or *VWF*, a higher expression level was associated with an increased risk of recurrence (Figure 3d, *p* < 0.05).

For each patient, a risk score or a CRPCPS score was constructed based on the multiplication of a gene’s expression level with its corresponding regression coefficient derived from the multivariate Cox regression model, i.e., CRPCPS score = (0.5932 × *NPEPL1* expression level) + (0.4505 × *VWF* expression level) + (−0.3704 × *LMO7* expression level) + (−0.3399 × *ALDH2* expression level) + (0.3545 × *NUAK1* expression level) + (−0.6515 × *TPT1* expression level). In the TCGA training cohort, the patients were stratified into high- or low-risk groups using the best cut-off value (Figure 4a). The number of recurrent patients increased with the rise of the CRPCPS score (Figure 4b). Consistently, the expression levels of *NPEPL1*, *VWF*, and *NUAK1* were upregulated, and the expression levels of *LMO7*, *ALDH2*, and *TPT1* were downregulated (Figure 4c). Kaplan–Meier analysis demonstrated that patients in the low-risk group had prolonged RFS compared to those in the high-risk group (*p* < 0.0001) (Figure 4d). The training cohort’s calibration plot showed an excellent agreement of 3-, 5-, and 8-year RFS probabilities between the predicted outcomes and actual observations (Figure 4e). The predictive accuracy of the CRPCPS was further evaluated by the time-dependent ROC in this cohort, with 3-, 5-, and 8-year AUCs of 0.769, 0.736, and 0.764 respectively, which were significantly larger than those of Gleason scores and pathological tumor stages (Figure 4f). The C-index was 0.743 (95% confidence interval (CI): 0.639, 0.847) in the training cohort, and the optimism-corrected C statistic with 1000 bootstrap replications was also 0.743.

### 3.3. Development of a CRPC Diagnostic Model Based on the CRPCPS Genes

To investigate the potential use of the six genes for predicting CRPC occurrence, the area under the receiver operating characteristic (ROC) curve (AUC) was applied to evaluate each of the six genes’ diagnostic effectiveness. In the training cohort, five in six genes (*NPEPL1*, *LMO7*, *VWF*, *NUAK1*, and *TPT1*) could efficiently distinguish CRPC from HSPC samples with an AUC value greater than 0.7 (Appendix A), four of which demonstrated a promising discrimination performance in the validation cohort (Appendix A), and these findings were consistent with their aberrant expression in CRPC samples (Figure 5a).

In addition, a logistic regression model was fitted to build the CRPC-diagnosis score in the training cohort after overcoming the data imbalance problem by the synthetic minority over-sampling technique (SMOTE), and the forward stepwise selection further optimized the model to a 3-gene panel (*NPEPL1*, *LMO7*, and *TPT1*) (Appendix A). This 3-gene CRPC diagnostic model showed high accuracy in distinguishing CRPC from HSPC samples in both the training and validation cohorts, with AUC values of 0.96 and 0.93, respectively (Figure 5b). Although the diagnostic model’s calibration curves for CRPC prediction in the training and validation cohorts displayed some departure between prediction and observation (Figure 5c), the CRPCPS showed low Brier scores (0.03 and 0.09 for the training and validation cohorts, respectively), and the Spiegelhalter *z* test yielded a nonsignificant statistic (*p* = 0.86 and 0.76 for the training and validation cohorts, respectively).

### 3.4. Validation of the CRPCPS in Independent Patient Cohorts

The prognostic value of the CRPCPS was first validated in the internal validation cohort, which contained 76 patients with a high risk and 55 patients with a low risk, defined by the CRPCPS scores (Appendix A). An increase in the CRPCPS score correlated with the upregulation of *NPEPL1*, *NUAK1*, and *VWF* but the downregulation of *ALDH2*, *LMO7*, and *TPT1* (Appendix A). The findings are also consistent with those from the training cohort, as patients in the high-risk group had a higher possibility of relapse than those in the low-risk group (*p* = 0.0038, Figure 6a, left panel). Although the CRPCPS’ AUCs are smaller than those of the Gleason score, they still displayed good discrimination in the internal validation cohort (all were greater than 0.65, Appendix A). The same conclusion remained valid in the entire TCGA cohort, which combined the training cohort and the internal validation cohort (Figure 6b, left panel, and Appendix A). AUC values of the CRPCPS at 3, 5, and 8 years in the entire cohort were similar to those in the training cohort (Appendix A). The 3-, 5-, and 8-year calibration plots of the internal validation cohort and the entire TCGA cohort resulted in a satisfactory concordance between the predicted and observed RFS probabilities (Figure 6a,b, right panel). Additionally, the C-index of the CRPCPS reached 0.70 with 1000 bootstrap iterations (95% CI: 0.56, 0.83) in the internal validation cohort and 0.73 (95% CI: 0.65, 0.82) in the entire TCGA cohort.

We also validated the CRPCPS in 6 completely independent datasets (Table 1). Again, patients in the low-risk group defined by the CRPCPS score had significantly longer RFS than those in the high-risk group in five of the six cohorts. Patients in these cohorts had received different therapies, including definitive radiotherapy in the Belfast cohort (GSE116918, *n* = 248, *p* = 0.009, Figure 6c), and radical prostatectomy in the MSKCC cohort (*n* = 140, *p* < 0.0001, Figure 6d), the Stockholm cohort (GSE70769, *n* = 92, *p* < 0.0001, Figure 6e), and the GSE54460 cohort (*n* = 91, *p* = 0.018, Figure 6f). The CPC-GENE cohort (GSE84042, *n* = 73) only had expression data for five of the six genes (*VWF*, *LMO7*, *ALDH2*, *NUAK1*, and *TPT1*), but patients of high- and low-risk groups defined by the five genes’ CRPCPS scores still had significantly different RFS (*p* = 0.025, Figure 6g). In the Cambridge cohort (GSE70768, *n* = 106), the prognostic significance of CRPCPS was borderline (*p* = 0.057, Figure 6h). The calibration plots from the six external validation cohorts also indicated a good agreement between the estimations and the observations (Figure 6). In addition, the C-index was 0.68 (95% CI: 0.57, 0.78) in the MSKCC cohort, 0.66 (95% CI: 0.57, 0.75) in the Stockholm cohort (GSE70769), and 0.62 (95% CI: 0.49, 0.75) in the Cambridge cohort. Less ideal C-index was observed in the Belfast cohort (C-index: 0.59; 95% CI, 0.52–0.67), the GSE54460 cohort (C-index: 0.58; 95% CI, 0.50–0.67), and the CPC-GENE cohort (C-index: 0.55; 95% CI, 0.44–0.74).

Overall survival (OS) data and metastasis-free survival (MFS) data were available for three datasets (MSKCC, Belfast, and GSE16560 [56]). We evaluated whether the CRPCPS predicts OS and/or MFS. As showed in Figure 7, the high-risk patients had a shorter OS than those in the low-risk group in both the MSKCC cohort (*p* = 0.01, Figure 7a) and the GSE16560 cohort (*p* = 0.052, Figure 7b). Only the Belfast cohort had available MFS data, and the CRPCPS also showed an excellent predictive ability for MFS (*p* < 0.0001, Figure 7c).

### 3.5. The CRPCPS Is Prognostic of RFS in PCa

The univariate analysis revealed that the CRPCPS was significantly associated with RFS in the training cohort (Figure 8a, upper panel), internal validation cohort (Figure 8b, upper panel), and the entire TCGA cohort (Figure 8c, upper panel), with hazard ratios (HRs) of 2.72, 2.18, and 2.51, respectively. The HRs of the CRPCPS scores for the remaining 6 cohorts ranged from 1.66 to 3.08 (*p* < 0.05 in all but one cohort). Multivariate Cox regression analysis further demonstrated that the signature remained as an independent prognostic factor after adjusting for various clinicopathologic characteristics, including age, Gleason score, tumor stage, and lymph node status in the training cohort (Figure 8a, lower panel). In all 9 cohorts, the CRPCPS was significant in 5 of them when variables with a sample size smaller than 20 were excluded for analysis, including the training cohort, the entire TCGA cohort, the MSKCC cohort, the Stockholm cohort (GSE70769), and the GSE54460 cohort (Figure 8a,c and Table 3). When all seven variables, when available, were included in the multivariate analysis, the CRPCPS was significant in three of the nine cohorts (i.e., the training cohort, the entire TCGA cohort, and the GSE54460 cohort).

### 3.6. The CRPCPS Remains Prognostic in PCa Patients with a Specific Clinicopathological Parameter

Six of the nine cohorts used in the CRPCPS’s discovery and validation analyses had greater numbers of patients. Thus, they were used for evaluating the association between the CRPCPS and various clinicopathological characteristics. The cohorts included the TCGA training, internal validation cohort, the entire TCGA cohort, and three validation cohorts (MSKCC, Cambridge, and Belfast cohort). The characteristics included tumor stage, lymph node status, Gleason score, and age at diagnosis. 

As expected, a higher CRPCPS score was positively associated with multiple adverse clinicopathological features, including higher tumor stage (Figure 9a and Appendix A), higher Gleason score (Figure 9b, left panel and Appendix A), and lymph node metastasis (Figure 9c and Appendix A). The exception was patients’ age (> vs. ≤65 years), which was not significantly associated with the CRPCPS score in any of the six cohorts (Appendix A).

Furthermore, the prognostic value of the CRPCPS was evaluated in patients stratified by tumor stage, lymph node status, and Gleason score in different cohorts. For such evaluations, we only used the subgroups that had 50 or more patients after stratification. Interestingly, a higher CRPCPS score was significantly associated with worse RFS in T1/T2 subgroups from three of the five cohorts (Figure 10a, left panel, and Appendix A). However, the *p*-values in the MSKCC cohort (Figure 10a, right panel, *p* = 0.16) and the Belfast cohort (GSE116918, Appendix A, *p* = 0.2) did not reach prognostic significance. The association between higher CRPCPS score and worse RFS was also detectable in patients with T3/T4 tumors in the training cohort (Figure 10b, left panel, and Appendix A), the entire TCGA cohort (Appendix A), the MSKCC cohort (Figure 10b, right panel), the Cambridge cohort (GSE70768, Appendix A), and the Belfast cohort (GSE116918, Appendix A). The internal validation cohort was the only one not showing a significant prognosis (*n* = 51, Appendix A, *p* = 0.1).

Further stratified analyses revealed that CRPCPS score could distinguish different prognoses in patients with a Gleason score of 7 or smaller (≤7) from 5 of 6 cohorts (Figure 10a,b and Appendix A), although a borderline prognostic significance was found in the Cambridge cohort (GSE70768, Appendix A, *p* = 0.07). Interestingly, such an association was also observed in patients with a Gleason score of 8 or above in multiple cohorts (Figure 10d and Appendix A).

Still, patients without lymph node metastasis (N0) in four of the five available cohorts showed disparate RFS between high- and low-risk groups stratified by CRPCPS score (Figure 10e and Appendix A). At the same time, it was not evident in the Cambridge cohort (GSE70768, Appendix A, *p* = 0.18). The number of patients with lymph node metastasis (N1) was too small for analysis in the entire TCGA cohort (Appendix A).

### 3.7. Model Comparison

To further evaluate this 6-gene signature’s predictive performance, we compared the CRPCPS with three published signatures in the entire TCGA cohort (*n* = 436) and the MSKCC cohort (*n* = 140), two large cohorts that had been used in at least two of the three published studies. In the entire TCGA cohort, the CRPCPS’ AUCs at 3, 5, and 8 years were 0.77, 0.74, and 0.73 respectively, which were greater than those in the Wang study [46] (0.64, 0.59, and 0.60, respectively), the Shi study [45] (0.66, 0.58, and 0.62, respectively), and the Hu study [20] (0.70, 0.57, and 0.64, respectively) (Appendix A, left panel, Wilcoxon *p* < 0.001). In addition, the CRPCPS had a smaller predicted error than the three signatures in the predicted error curves (Appendix A, left panel). In the MSKCC cohort, while the Shi signature performed better than the CRPCPS, the CRPCPS performed better than those of the Wang and Hu signatures (Appendix A, right panel).

### 3.8. Nomogram Based on CRPCPS Scores and Clinicopathological Features

To establish a quantitative approach for RFS prediction, we incorporated CRPCPS score with other clinicopathological features (i.e., age at diagnosis, Gleason score, pathological tumor stage, and lymph node status) to construct a nomogram in the TCGA training cohort. A point was calculated for each of the factors, and the total points of all factors were then obtained for the estimation of RFS rates at 3, 5, and 8 years (Figure 11a). 

The ROC curve and calibration plot were then drawn to evaluate the reliability of the nomogram. The AUCs for predicting RFS at 3, 5, and 8 years were 0.808, 0.757, and 0.822 respectively, in the training cohort (Figure 11b). In the internal validation cohort, the AUCs were only available for 3 and 5 years, and both were smaller than those in the training cohort (0.686 and 0.69, respectively) (Figure 11c). In the entire TCGA cohort, the AUCs were 0.774, 0.74, and 0.742 respectively, for the three time points (Figure 11d). The C-index of our nomogram with 1000 bootstrap resamples reached 0.71 (95% CI: 0.69–0.88), 0.77 (95% CI: 0.74–0.91), and 0.76 (95% CI: 0.70–0.85) in the training cohort, internal validation cohort, and the entire TCGA cohort, respectively. In addition, the line-segment in the calibration plots was close to the 45° line, indicating an excellent agreement between the prediction and observation in the training cohort (Figure 11e), the internal validation cohort (Figure 11f), and the entire TCGA cohort (Figure 11g).

### 3.9. Functional Enrichment Analysis of the CRPCPS Genes

The GO enrichment and KEGG pathway analyses were performed to explore the biological processes and pathways associated with the CRPCPS using the gene set variation analysis (GSVA). Patients in the training cohort were divided into high- or low-risk groups with the CRPCPS scores. Biological processes enriched in the high-risk group included cell cycle, heterochromatin organization, Ras signal transduction pathway, and macrophage differentiation (Appendix A). Those enriched in the low-risk group included metabolic process, androgen biosynthetic and metabolic process, and response to extracellular stimuli (Appendix A). 

The pathways of cell cycle regulation, ABC transporters, Hedgehog signaling, Toll-like receptor signaling, and MAPK (Mitogen-activated protein kinase) signaling were enriched in the high-risk group (Appendix A). In contrast, those of glycolysis gluconeogenesis, autophagy, and steroid hormone biosynthesis were enriched in the low-risk group (Appendix A).

## 4. Discussion

This study applied an integrated approach to publicly available PCa expression datasets to identify CRPC-associated genes and develop expression-based molecular signatures to predict RFS and CRPC. Associations of developed signatures with the overall survival, metastasis-free survival, and clinicopathological characteristics of PCa were also evaluated. In addition, we systematically assessed the prognostic value of the CRPCPS in PCa patients with a specific clinicopathological parameter. Another unique aspect of this study is that both CRPC and HSPC samples used for DEGs analysis were from the primary site (i.e., the prostate), minimizing issues related to CRPC samples from metastases because metastasis is accompanied by many genes’ expression changes [57].

Many genes’ expression changes are associated with CRPC. We found 287 genes that were differentially expressed between localized CRPC and HSPC samples, and 207 of them (Appendix A) were not discovered in previous studies with the same type of tumor specimens [28,29]. Many cellular processes and molecular pathways involving the 287 DEGs have been described in previous studies, including the extracellular matrix organization, PI3K-Akt, and ERK signaling pathways, and hormonal signaling pathways [29,58,59,60], as revealed by the GO enrichment and KEGG analysis (Figure 2).

With as few as six genes, the CRPCPS can predict RFS in PCa patients. Twenty-four of the 287 CRPC-associated genes showed a significant correlation between their expression change and RFS, according to the univariate Cox analysis (Appendix A). Combined LASSO and the forward stepwise selection analyses of the 24 genes resulted in the most robust model with the fewest genes capable of predicting RFS, i.e., a panel of 6 genes significantly correlates with RFS (Figure 3 and Figure 4, Table 2). This six-gene panel was named CRPC-derived prognosis signature (CRPCPS), and the CRPCPS score is based on the six genes’ expression levels and regression coefficients. The model was validated in eight patient cohorts, comprising more than 1300 patients (Figure 6). 

Interestingly, three of the six CRPCPS genes can effectively discriminate CRPC patients from HSPC individuals based on their expression levels and regression coefficients, as demonstrated by the stepwise selection analysis (Figure 5). It should be mentioned that the CRPCPS is likely not as effective as expected in the prediction of CRPC patients’ OS because of limited cohort sizes (*n* = 106, SU2C/PCF (Stand up to cancer/Prostate cancer foundation) Dream Team cohort and GSE35988 cohort). More CRPC cohorts needed to be included for evaluating the diagnostic value of the 3-gene model and the CRPCPS’ prognostic value for CRPC patients.

Our 6-gene CRPCPS model showed a reasonable performance in both calibration and discrimination. In addition to CRPCPS’s validity in all eight cohorts of more than 1300 patients, the C-index was acceptable in half of the eight cohorts, suggesting an acceptable prediction capability. The estimations and the actual observations had acceptable agreements in all the eight cohorts at almost all time points in the calibration plots (Figure 6). As expected, the C-index of the nomogram comprising the CRPCPS, patient’s age at diagnosis, Gleason score, tumor stage, and lymph node status showed better discrimination than CRPCPS alone. Therefore, the CRPCPS model performs equally well in calibration and discrimination.

The CRPCPS also appears to be a prognostic factor for MFS and OS, as it could effectively divide patients into subgroups with disparate MFS and OS. Although this conclusion needs further validation in additional datasets, the CRPCPS is associated with recurrence and metastasis (Figure 7).

The CRPCPS is prognostic of RFS in PCa patients and can predict RFS in PCa patients with a specific clinicopathological parameter. For example, multivariate analysis demonstrated that the CRPCPS score was an effective risk factor for RFS prediction independent of multiple known prognostic factors in five of nine PCa cohorts (Figure 8 and Table 3). Multivariate analyses of RFS further demonstrated that the CRPCPS’ hazard ratios (HRs) were 2.72 (95% CI: 1.66, 4.48) in the training cohort and 2.03 (95% CI: 1.41, 2.92) in the entire TCGA cohort (Figure 8). Similar HRs were found in three of the seven additional validation cohorts, including the MSKCC cohort (HR = 2.15; 95% CI: 1.30–3.56), the Stockholm cohort (HR = 2.68; 95% CI: 1.34–5.37), and the GSE54460 cohort (HR = 1.74; 95% CI: 1.07–2.84)). These findings suggest that the CRPCPS is capable of RFS prediction in PCa. The CRPCPS score was significantly correlated with the Gleason score, tumor stages, and lymph node metastasis (Figure 9). In addition, the CRPCPS remained prognostic when patients were stratified by different Gleason score, tumor stage, or lymph node status (Figure 10 and Appendix A). Furthermore, an enhanced predictive capability was detected after integrating the CRPCPS with the Gleason score, tumor stage, lymph node status, and age at diagnosis into a nomogram (Figure 11).

The performance of this 6-gene CRPCPS was improved when compared to some previously reported models. For example, the CRPCPS showed a more promising discrimination capability than two of the three published transcriptomic signatures, consisting of 3, 9, or 22 genes [20,45,46] (Appendix A). We noticed that none of the genes from the three published signatures [20,45,46] were in the 287 CRPC-associated genes identified in our study (Appendix A). One of the published signatures was entirely based on genes implicated in the autophagy process [20], whereas the two others were based on genes’ association with the Gleason score and biochemical recurrence-free survival status.

The biological mechanisms underlying the CRPCPS’s prognostic value appear to involve multiple established oncogenic signaling pathways. After patients were divided into the high- and low-score groups according to their CRPCPS scores and the GSVA analysis was applied, the most enriched biological processes or signaling pathways in the CRPCPS-high group are mostly oncogenic. For example, the processes of cell cycle progression, stem cell division, and macrophage were enriched (Appendix A). Interestingly, androgen biosynthetic and fatty acid metabolic processes were highly enriched in the low-risk groups (Appendix A).

The pathways of Hedgehog signaling, MAPK signaling, Toll-like receptor signaling, and cell adhesion signaling were also enriched in the high-risk group. Previous studies have demonstrated that Hedgehog signaling and MAPK signaling are activated in more advanced localized PCa or enzalutamide-resistant PCa [61,62,63]. The Toll-like receptor and the cell adhesion signaling pathways have also been implicated in CRPC [64,65]. Adhesion molecules are well-known for their role in the epithelial-to-mesenchymal transition, which is considered a critical step in PCa disease progression. These signaling pathways’ activation in the high-risk group may help explain why patients in the high-risk group have shorter RFS than those in the low-risk group. 

Regarding the six genes that constitute the CRPCPS, including *VWF*, *LMO7*, *ALDH2*, *NPEPL1*, *NUAK1*, and *TPT1*, two of them have been implicated in PCa. One is *ALDH2* (aldehyde dehydrogenase 2), a stem cell marker that promotes cell proliferation and doxorubicin resistance when overexpressed in cancer cells [66]. *ALDH2*’s expression is associated with OS in lung adenocarcinoma, hepatocellular carcinoma, and upper tract urothelial carcinoma [67,68,69,70]. In PCa, *ALDH2* is expressed at a lower level in CRPC than localized tumors, and patients with higher *ALDH2* expression have a better prognosis [71], which is consistent with our finding in the current study. Another gene is *TPT1*, a direct target gene of p53 with elevated expression in malignancies, including PCa, glioma, and skin squamous cell carcinoma [72]. It functions in many cellular processes, including cell growth, cell cycle progression, malignant transformation, and apoptosis. As an androgen-regulated gene [73], *TPT1* is often upregulated in CRPC. *TPT1*’s upregulation correlates with CRPC progression [74] and higher PCa grades [75], similar to our finding. 

At least three other genes have been well implicated in cancer development and progression. The Von Willebrand Factor (*VWF*) is a marker for angiogenesis, and it has been suggested as a predictive biomarker for therapeutic response in pancreatic cancer [76]. A decrease in *VWF* expression is associated with improved survival in patients with colorectal cancer and liver metastases [77], and a single nucleotide polymorphism in VWF (rs73049469) is associated with overall survival in NSCLC [78]. *VWF* is also upregulated in androgen-independent prostate cancer cells [79]. 

*LMO7* participates in cytoskeletal reorganization during carcinogenesis [80,81,82], and its downregulation in lung cancer is associated with poorer patient prognosis [83]. Its absence in mice induces spontaneous lung adenocarcinomas [84]. Although not reported in prostate cancer, reduced *LMO7* expression inhibits breast cancer cell migration [85], and the *LMO7*-*BRAF* fusion frequently occurs in papillary thyroid cancer [86]. 

*NUAK1*, also known as *ARK5*, encodes for an AMPK-related serine/threonine protein kinase that functions in cell adhesion [87], migration [88,89], and cellular and organismal metabolism [90]. It is overexpressed in multiple types of cancers, and the overexpression is associated with poorer prognosis in breast cancer [91], hepatocellular carcinoma [92], colorectal cancer [93,94], and lung cancer [89]. A role in PCa has not been reported for *NUAK1*, though. 

It is unclear whether *NPEPL1* plays a role in PCa and other cancers. However, it appears to be a direct target gene of miR-19a, an oncogenic miRNA in breast cancer [95], and the *STX16*-*NPEPL1* fusion event occurs in gastrointestinal stromal tumors [96]. These six genes constitute the CRPCPS, suggesting that they may play a coordinated role in CRPC progression and are worthy of further investigation. 

While the novel six-gene CRPCPS has reasonable generalizability and versatility in RFS prediction and the three-gene diagnostic model can distinguish CRPC from HSPC, we are aware that our study has some limitations. Firstly, the sample size of CRPC could be an issue, as a relatively small number of CRPCs were used to identify CRPC-associated genes and assess the diagnostic model. Secondly, the prognostic performance of this six-gene signature needs to be evaluated in clinical practice. Another limitation derives from the heterogeneities among the patient cohorts used in our study. For example, the follow-up time was shorter in the Cambridge cohort (GSE70768, median 29.7 months) than in the Belfast cohort (GSE116918, median 82 months). No standardization was conducted in terms of sample collection, RNA extraction, gene expression analysis, etc. These heterogeneities made us choose different optimal thresholds for risk stratification.

## 5. Conclusions

In summary, this study identified 287 genes differentially expressed between localized CRPC and HSPC samples, many of which were not reported in previous studies. A six-gene panel derived from the 287 CRPC-associated genes can predict RFS in PCa patients, including those with lower tumor stages or lower Gleason scores. Three of the six genes comprised of a diagnostic model predictive of CRPC. Therefore, these gene signatures may help develop CRPC and RFS biomarkers via large-scale randomized clinical trials.

## Figures and Tables

**Figure 1 cancers-13-00917-f001:**
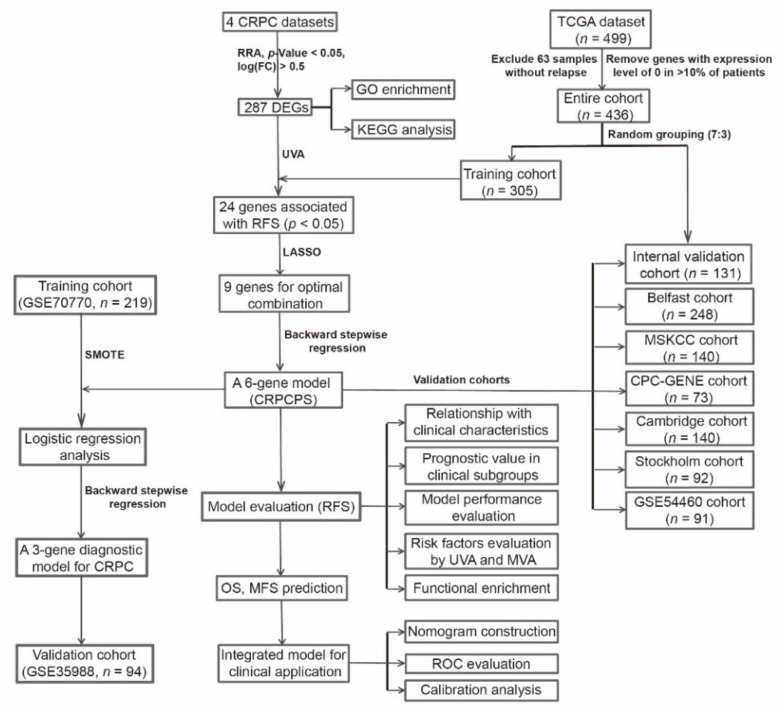
The workflow of the study. CRPC: castration-resistant prostate cancer; RRA: Robust Rank Aggregation; LASSO: Least absolute shrinkage and selection operator; DEGs: differentially expressed genes; UVA: Univariate Cox regression analysis; GO: Gene Ontology; KEGG: Kyoto Encyclopedia of Genes and Genomes; RFS: recurrence-free survival; CRPCPS: CRPC-derived prognostic signature; GSVA: Gene set variation analysis; SMOTE: synthetic minority over-sampling technique; ROC: receiver operating characteristic.

**Figure 2 cancers-13-00917-f002:**
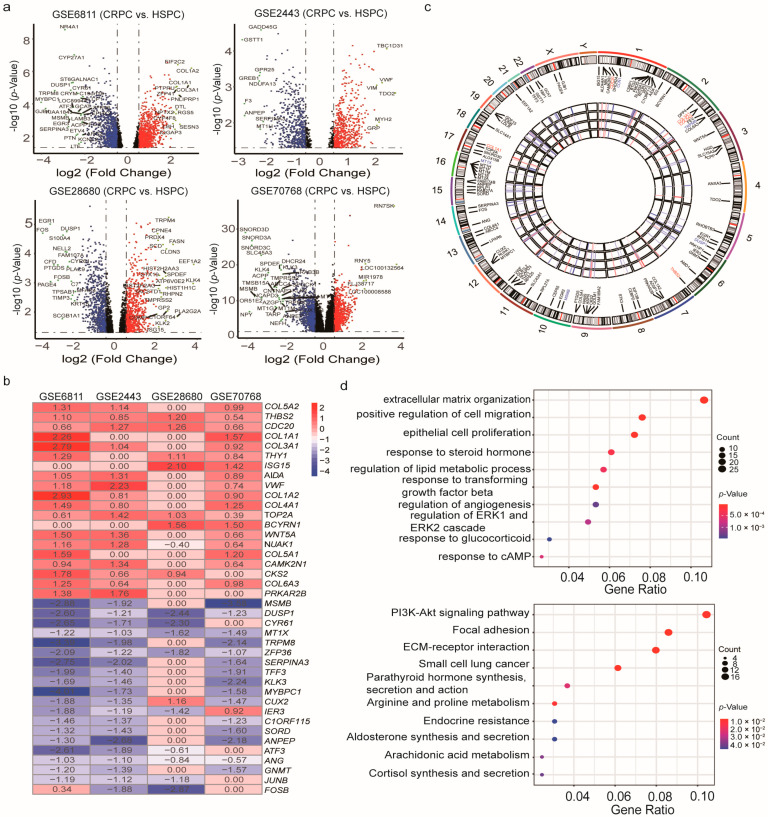
Identification of differentially expressed genes (DEGs) between localized CRPC and HSPC samples from 4 unrelated cohorts. (**a**) Volcano plots of datasets GSE6811 and GSE2443 (upper panel), and GSE28680 and GSE70768 (lower panel) from the GEO database. DEGs with |log2 (Fold change)| > 2 would be listed, additionally. (**b**) Heatmap of the top 20 downregulated and the top 20 upregulated genes from an integrated analysis of 4 datasets. Each column represents a dataset and each row a gene. The number in each rectangle represents the value of log2 (fold change). The gradual color ranging from blue to red represents the changing process from down- to up-regulation (CRPC vs. HSPC). (**c**) Circular visualization of expression patterns and chromosomal positions of the top 100 DEGs. The 4 prostate cancer (PCa) microarray datasets from the GEO database are represented in the inner circular heatmaps. Red, blue, and white indicate upregulated, downregulated, and absent genes in a given dataset, respectively. The outer circle represents chromosomes, and lines coming from each gene point to their specific chromosomal locations. The top five upregulated and downregulated genes ordered by *p*-value were marked in red and blue, respectively. (**d**) GO enrichment (upper panel) and KEGG pathway (lower panel) analyses of differentially expressed genes (DEGs) between localized CRPC and HSPC samples. CRPC: Castration-resistant prostate cancer; HSPC: Hormone-sensitive prostate cancer; GO: Gene Ontology; KEGG: Kyoto Encyclopedia of Genes and Genomes. The size of each dot represents the count of genes, and the color represents the *p*-value.

**Figure 3 cancers-13-00917-f003:**
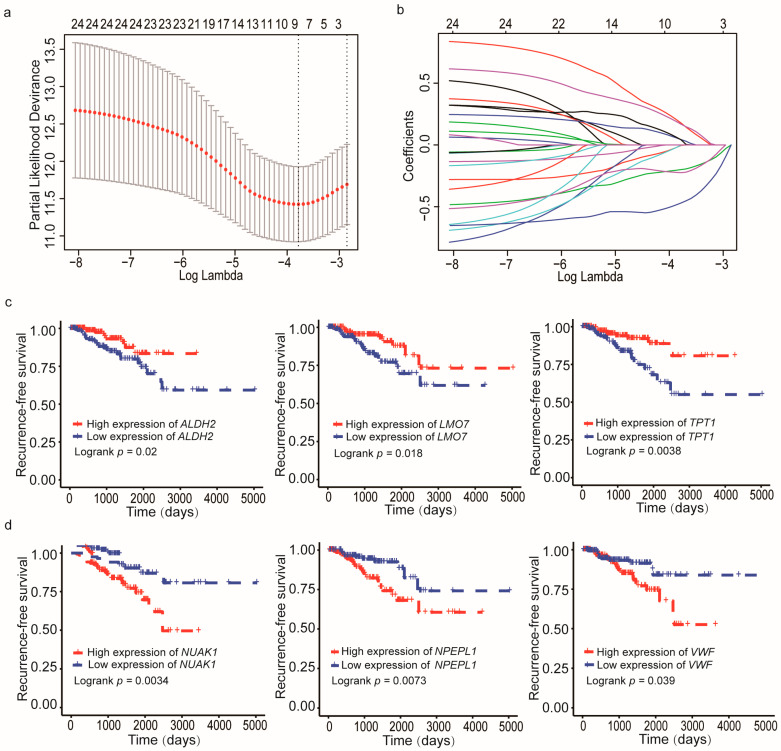
Identification of CRPC-associated genes that correlate with recurrence-free survival (RFS) in PCa. (**a**) Partial likelihood deviance of different numbers of variables revealed by the LASSO regression model. The red dots represent the partial likelihood deviance values. The grey lines represent the partial likelihood deviance ± standard error (SE). The two vertical lines on the left and right represent optimal values by minimum criteria and 1-SE criteria. The proper log (Lambda) value was chosen by 10-fold cross-validation via minimum criteria. LASSO: least absolute shrinkage and selection operator method. (**b**) LASSO coefficient profiles of the 24 CRPC-associated genes. (**c**) The Kaplan–Meier survival curves showing patients with higher expression levels of *ALDH2* (left), *LMO7* (center), and *TPT1* (right) had a favorable prognosis in PCa (*p* < 0.05). (**d**) The Kaplan–Meier curves showing higher expression levels of *NUAK1* (left), *NPEPL1* (center), and *VWF* (right) indicated poor prognosis in PCa (*p*-value < 0.05).

**Figure 4 cancers-13-00917-f004:**
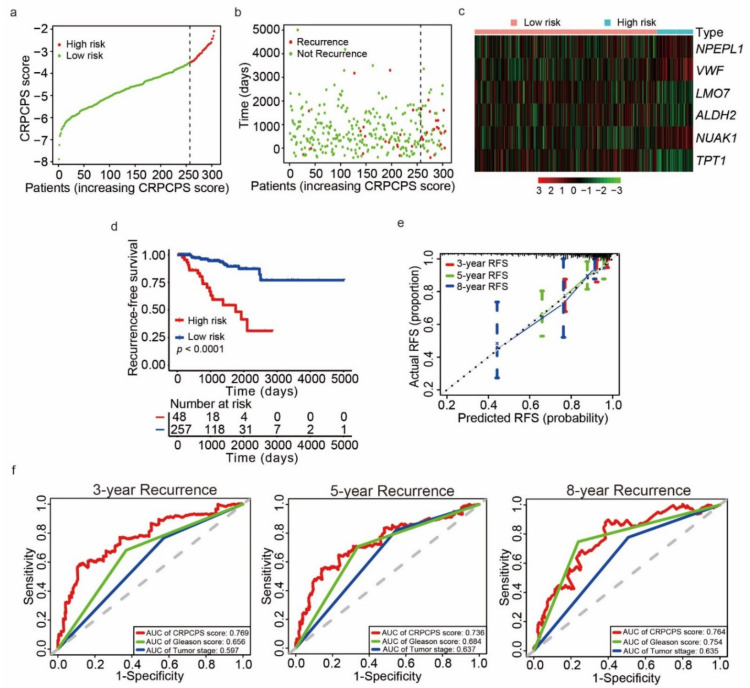
Construction of the CRPC-derived prognosis signature (CRPCPS) in the TCGA training cohort for predicting RFS in PCa. (**a**–**c**) Distribution of CRPCPS score, patients’ recurrent status, and expression pattern of the six CRPCPS genes in the TCGA training cohort, respectively. (**d**) Kaplan–Meier curves of recurrence-free survival (RFS) according to CRPCPS score in the TCGA training cohort. (**e**) Calibration curves of the CRPCPS for predicting RFS probability at 3, 5, and 8 years in the TCGA training cohort. (**f**) Receiver operating characteristic (ROC) curves of the CRPCPS score, Gleason score, and pathological tumor stage were performed to evaluate the predictability of RFS at 3 (left), 5 (center), and 8 years (right) in the TCGA training cohort.

**Figure 5 cancers-13-00917-f005:**
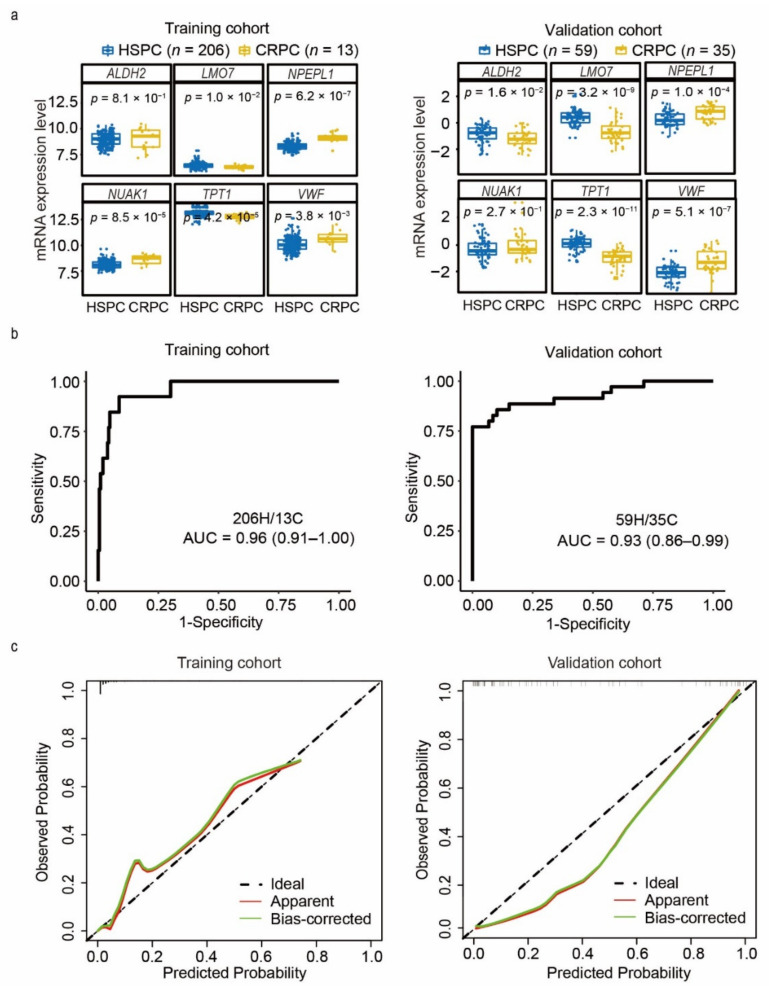
Three-gene panel derived from the six CRPCPS genes for prediction of CRPC occurrence. (**a**) Box plots of the six CRPCPS genes’ expression value in the training cohort (left panel) and validation cohort (right panel). ROC curves (**b**) and calibration plots (**c**) of the 3-gene CRPC diagnostic model in the training cohort (left panel) and validation cohort (right panel). H: HSPC, C: CRPC. The diagonal dotted line represents a perfect prediction by an ideal model.

**Figure 6 cancers-13-00917-f006:**
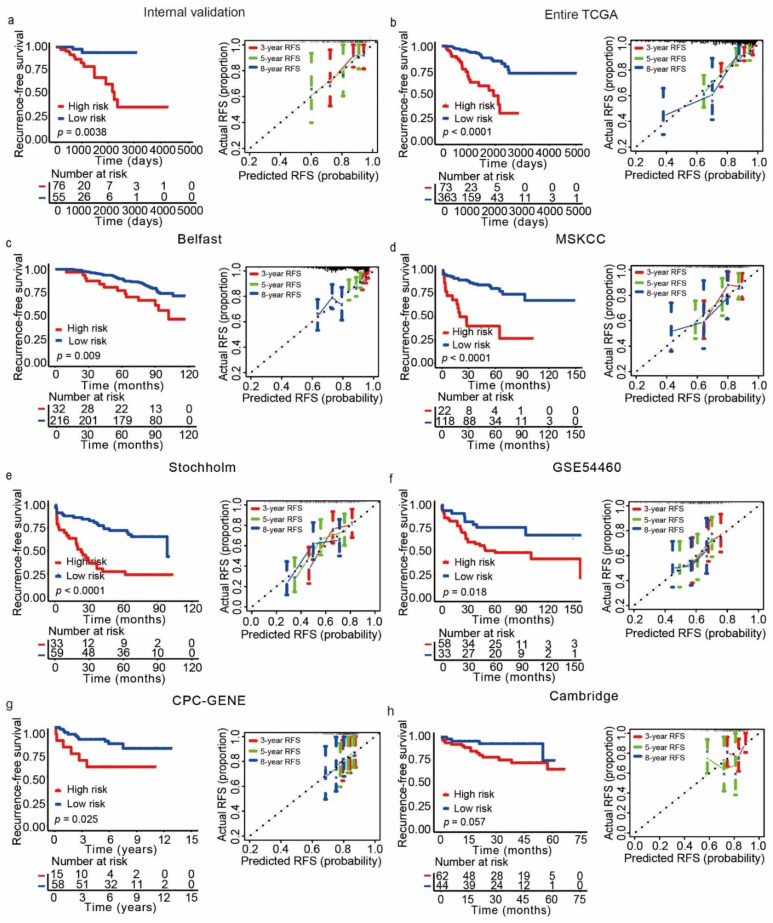
Performance of the CRPCPS in different validation cohorts. Kaplan–Meier plots were used to visualize the RFS probabilities for the high-risk versus low-risk patients (left panel). The calibration curves were applied to evaluate 3-, 5-, and 8-year RFS probabilities (right panel) in the internal validation cohort (**a**), the Entire TCGA cohort (**b**), the Belfast cohort (GSE116918) (**c**), the MSKCC cohort (**d**), the Stockholm cohort (GSE70769) (**e**), the GSE54460 cohort (**f**), the CPC-GENE cohort (**g**), and the Cambridge cohort (GSE70768) (**h**).

**Figure 7 cancers-13-00917-f007:**
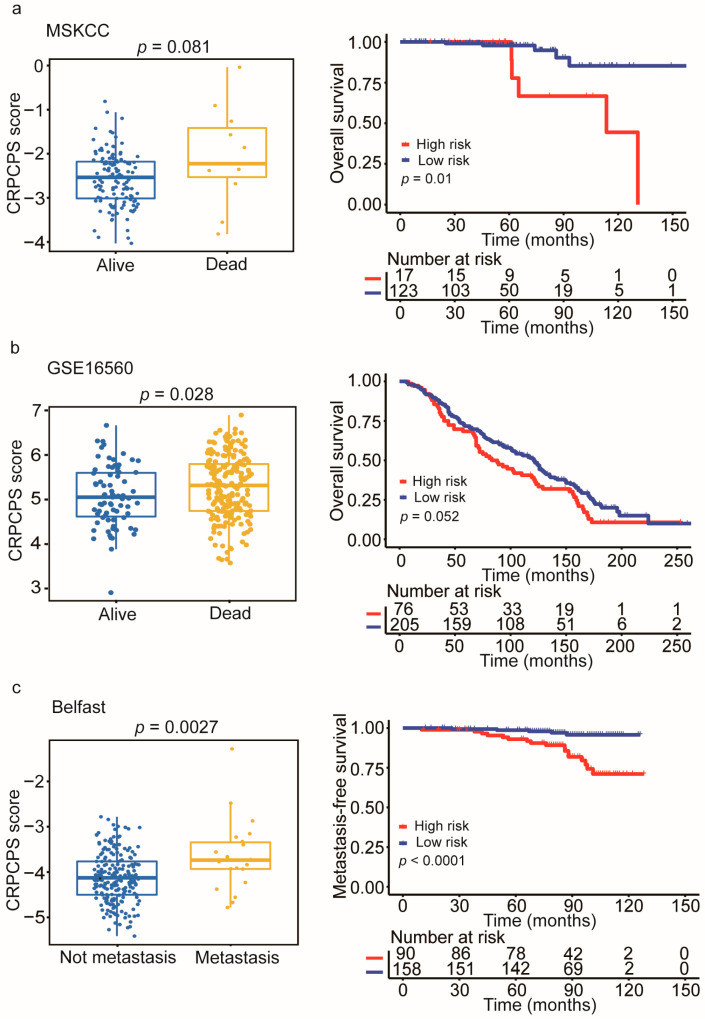
The association of CRPCPS with patients’ survival/metastatic status and CRPCPS’s performance in predicting overall survival (OS) and metastasis-free survival (MFS) in different patient cohorts. Association of CRPCPS with patients’ survival/metastatic status (left panel) and Kaplan–Meier plot (right panel) for predicting OS/MFS in the MSKCC cohort (left, *n* = 140) (**a**), the GSE16560 cohort (left, *n* = 281) (**b**), and the Belfast cohort (left, *n* = 248) (**c**). The differences between the two groups were determined by the log-rank test.

**Figure 8 cancers-13-00917-f008:**
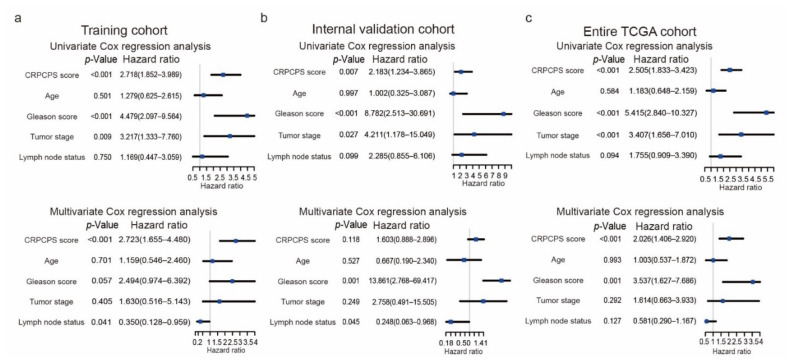
Univariate and multivariate Cox regression analyses of the CRPCPS score, patients’ age at diagnosis, Gleason score, tumor stage, and lymph node status in the TCGA training cohort (**a**), the internal validation cohort (**b**), and the entire TCGA cohort (**c**). The blue squares represent the hazard ratio (HR), and the black lines represent the 95% CI. CI, confidence interval. Age: age at diagnosis; Tumor stage: pathological tumor stage.

**Figure 9 cancers-13-00917-f009:**
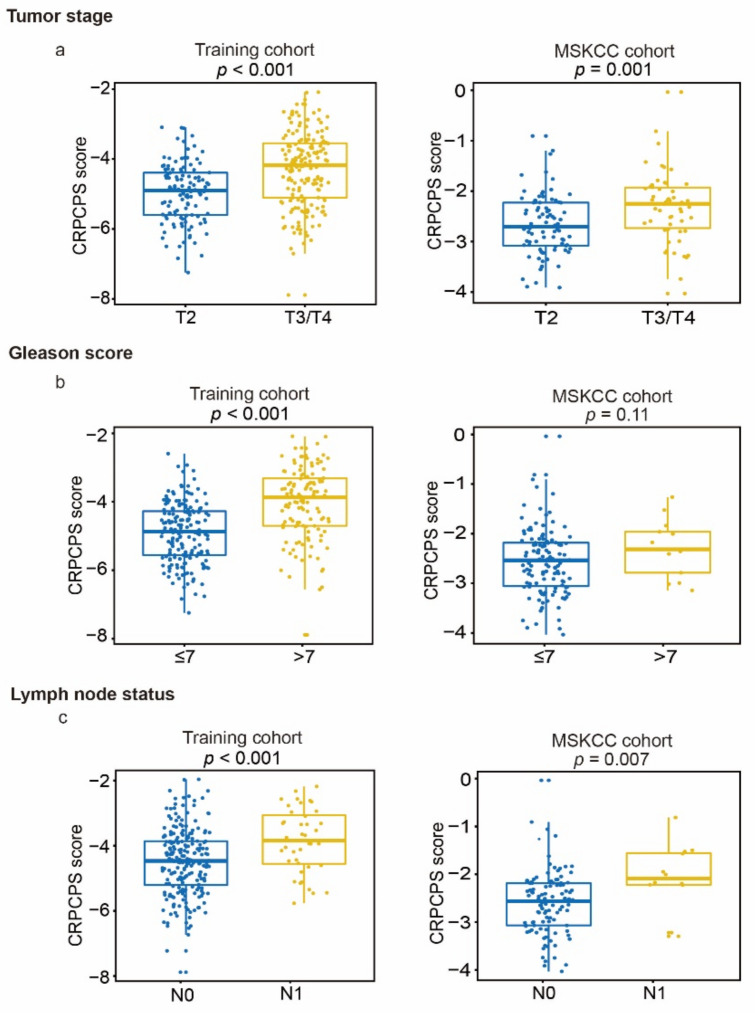
Associations of CRPCPS with tumor stage (**a**), Gleason score (**b**), and lymph node status (**c**) in the training cohort (left panel) and the MSKCC cohort (right panel).

**Figure 10 cancers-13-00917-f010:**
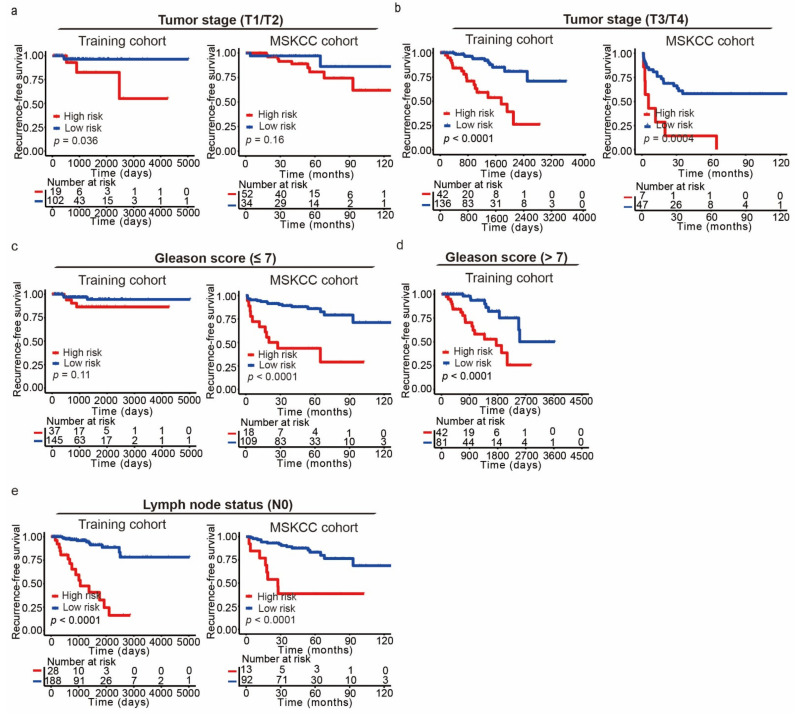
Kaplan–Meier curves of recurrence-free survival (RFS) according to the CRPCPS score in patients stratified by pathological tumor stage, lymph nodes status, and Gleason score. Kaplan–Meier curves were applied to patients with lower tumor stage (T1/T2) (**a**), higher tumor stage (T3/T4) (**b**), lower Gleason score (≤7) (**c**), higher Gleason score (>7) (**d**), and without lymph node metastasis (N0) (**e**), in the training cohort (left panel) and the MSKCC cohort (right panel). The tick marks on the Kaplan–Meier curves represent the censored subjects. The two-sided log-rank test was used to determine differences between the two curves.

**Figure 11 cancers-13-00917-f011:**
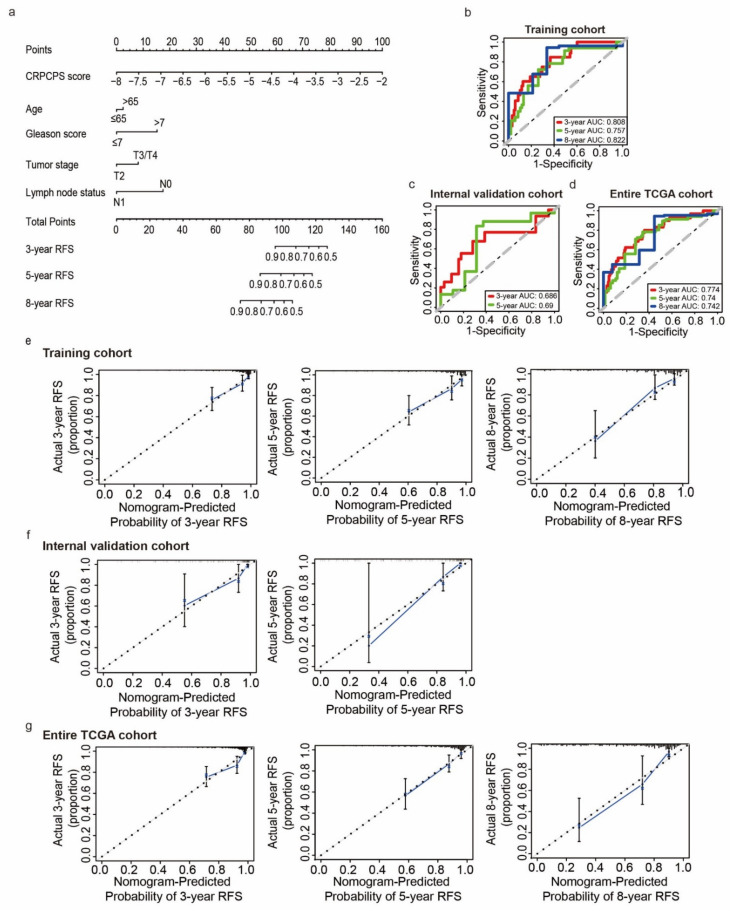
Nomogram construction and evaluation for predicting 3-, 5-, and 8-year RFS in PCa. (**a**) Nomogram construction for the 3-, 5-, and 8-year RFS probability prediction in the TCGA training cohort. ROC curves evaluated the efficiency of the nomogram for predicting 3-, 5-, and 8-year RFS in the TCGA training cohort (**b**), the internal validation cohort (**c**), and the entire TCGA cohort (**d**). Calibration plots of the nomogram for predicting the probability of RFS at 3 (left), 5 (center), and 8 years (right) in the TCGA training cohort (**e**), the internal validation cohort (**f**), and the entire TCGA cohort (**g**).

**Table 1 cancers-13-00917-t001:** Characteristics of patient cohorts used for the analysis of recurrence-free survival (RFS).

Characteristics	Training Cohort	Validation Cohort						
	TCGA (*n* = 305)	TCGA (*n* = 131)	Belfast (*n* = 248)	MSKCC (*n* = 140)	CPC-GENE (*n* = 73)	Cambridge (*n* = 106)	Stockholm (*n* = 92)	GSE54460 (*n* = 91)
Age (year) (%)								
≤65	216 (70.8)	92 (70.2)	87 (35.1)	117 (83.6)	57 (78.1)	81 (76.4)	NA	67 (73.6)
>65	89 (29.2)	39 (29.8)	161 (64.9)	23 (16.4)	16 (21.9)	25 (23.6)	NA	24 (26.4)
Median follow-up time (months)	30.8	25.9	82	45.5	71.3	29.7	58	59.5
Recurrent events (%)	35 (11.5)	17 (13.0)	56 (22.6)	36 (25.7)	16 (21.9)	19 (17.9)	45 (48.9)	40 (44.0)
Tumor stage (%)								
T1/T2	121 (39.7)	54 (41.2)	127 (51.2)	86 (61.4)	40 (54.8)	33 (31.1)	47 (51.1)	75 (82.4)
T3/T4	178 (58.4)	76 (58.0)	96 (38.7)	54 (38.6)	33 (45.2)	73 (68.9)	42 (45.7)	15 (16.5)
Not available	6 (2)	1 (0.8)	25 (10.1)	0 (0)	0 (0)	0 (0)	3 (3.3)	1 (1)
Lymph node status (%)								
N0	216 (70.8)	92 (70.2)	NA	105 (75)	NA	77 (72.6)	18 (19.6)	NA
N1	40 (13.1)	22 (16.8)	NA	12 (8.6)	NA	6 (5.7)	NA	NA
Not available	49 (16.1)	17 (13)	NA	23 (16.4)	NA	23 (21.7)	74 (80.4)	NA
Gleason score (%)								
≤7	182 (59.7)	80 (61.1)	141 (56.9)	127 (90.7)	73 (100)	97 (91.5)	75 (81.5)	80 (87.9)
>7	123 (40.3)	51 (38.9)	107 (43.1)	13 (9.3)	0 (0)	9 (8.5)	15 (16.3)	11 (12.1)
Not available	0 (0)	0 (0)	0 (0)	0 (0)	0 (0)	0 (0)	2 (2.2)	0 (0)
PSA level (ng/mL)								
≤10	249 (81.6)	102 (77.9)	50 (20.2)	114 (81.4)	57 (78.1)	79 (74.5)	62 (67.4)	61 (67)
>10	3 (1)	5 (3.8)	198 (79.8)	24 (17.1)	16 (21.9)	26 (24.5)	28 (30.4)	27 (29.7)
Not available	53 (17.4)	24 (18.3)	0 (0)	2 (1.4)	0 (0)	1 (1)	2 (2.2)	3 (3.3)
Radiation therapy (%)								
No	256 (83.9)	107 (81.7)	NA	NA	NA	NA	NA	NA
Yes	37 (12.1)	17 (13.0)	NA	NA	NA	NA	NA	NA
Not available	12 (3.9)	7 (5.3)	NA	NA	NA	NA	NA	NA
Targeted therapy (%)								
No	262 (85.9)	109 (83.2)	NA	NA	NA	NA	NA	NA
Yes	31 (10.2)	14 (10.7)	NA	NA	NA	NA	NA	NA
Not available	12 (3.9)	8 (6.2)	NA	NA	NA	NA	NA	NA
Surgical margins (%)								
Positive	NA	NA	NA	107 (76.4)	NA	83 (78.3)	50 (54.3)	54 (59.3)
Negative	NA	NA	NA	33 (23.6)	NA	23 (21.7)	42 (45.7)	32(35.2)
Not available	NA	NA	NA	0 (0)	NA	0 (0)	0 (0)	5 (5.5)

TCGA: The Cancer Genome Atlas; MSKCC: The Memorial Sloan Kettering Cancer Center; CPC-GENE: Canadian Prostate Cancer Genome Network; NA: not available; Age: age at diagnosis; Tumor stage: pathological tumor stage; N0: without lymph node metastasis; N1: with lymph node metastasis; PSA: prostate-specific antigen.

**Table 2 cancers-13-00917-t002:** Identification and chromosomal location of the six CRPCPS genes and their association with recurrence-free survival (RFS) in prostate cancer.

Entrez ID	Gene Symbol	Full Name	Location	HR	*p*-Value
79716	*NPEPL1*	Aminopeptidase Like 1	20q13.32	2.564	0.01
7450	*VWF*	Von Willebrand Factor	12p13.31	2.091	0.043
4008	*LMO7*	LIM Domain 7	13q22.2	0.433	0.022
217	*ALDH2*	Aldehyde Dehydrogenase 2 Family Member	12q24.12	0.416	0.024
9891	*NUAK1*	NUAK Family Kinase 1	12q23.3	2.884	0.005
7178	*TPT1*	Tumor Protein, Translationally-Controlled 1	13q14.13	0.354	0.006

CRPCPS: CRPC-derived prognostic signature; HR: Hazard Ratio.

**Table 3 cancers-13-00917-t003:** Multivariable analysis of the CRPCPS and clinicopathological characteristics in six additional independent datasets.

Study	Variable	Multivariable Analysis	
		HR	*p*-Value
Belfast	CRPCPS score	1.49 (0.923–2.408)	0.103
	Gleason score (>7 vs. ≤7)	1.077 (0.581–1.997)	0.814
	Age at diagnosis (>65 vs. ≤65)	0.911 (0.513–1.617)	0.749
	PSA level (>10 vs. ≤10)	1.202 (0.545–2.648)	0.648
	Tumor stage (T3/T4 vs. T1/T2)	1.807 (0.94–3.475)	0.076
MSKCC	CRPCPS score	2.150 (1.298–3.561)	0.003
	Surgical margins (Positive vs. Negative)	1.034 (0.491–2.176)	0.930
	Tumor stage (T3/T4 vs. T1/T2)	3.406 (1.516–7.653)	0.003
	PSA level (>10 vs. ≤10)	1.351 (0.616–2.963)	0.453
	Age at diagnosis (>65 vs. ≤65)	1.312 (0.562–3.060)	0.530
Stockholm	CRPCPS score	2.681 (1.339–5.371)	0.005
	Tumor stage (T3/T4 vs. T1/T2)	3.120 (1.567–6.218)	0.001
	PSA level (>10 vs. ≤10)	1.883 (0.989–3.586)	0.054
	Surgical margins (Positive vs. Negative)	1.568 (0.816–3.014)	0.178
GSE54460	CRPCPS score	1.742 (1.067–2.842)	0.026
	PSA level (>10 vs. ≤10)	3.115 (1.561–6.217)	0.001
	Age at diagnosis (>65 vs. ≤65)	0.919 (0.429–1.970)	0.828
	Surgical margins (Positive vs. Negative)	2.708 (1.335–5.493)	0.006
CPC-GENE	CRPCPS score	1.566 (0.618–3.971)	0.344
	Tumor stage (T3/T4 vs. T1/T2)	3.201 (1.105–9.271)	0.032
Cambridge	CRPCPS score	1.913 (0.821–4.460)	0.133
	Tumor stage (T3/T4 vs. T1/T2)	1.447 (0.460–4.556)	0.527
	PSA level (>10 vs. ≤10)	1.162 (0.403–3.351)	0.781
	Age at diagnosis (>65 vs. ≤65)	1.723 (0.660–4.500)	0.267
	Surgical margins (Positive vs. Negative)	1.339 (0.484–3.701)	0.574

CRPCPS: CRPC-derived prognostic signature; Tumor stage: pathological tumor stage; HR: hazard ratio.

## Data Availability

The data presented in this study are openly available in the Cancer Genome Atlas Program (TCGA), the cBio Cancer Genomics Portal (cBioportal), and Gene Expression Omnibus (GEO).

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
