# Peer review of "Novel Gene Signatures Predictive of Patient Recurrence-Free Survival and Castration Resistance in Prostate Cancer"

_cancers, 2021, doi:10.3390/cancers13040917_

Round 1
Reviewer 1 Report
In the manuscript entitled “Novel gene signatures predictive of patient recurrence-free survival and castration resistance in prostate cancer”, Dr. Dong’s team established a 6-gene signature, castration resistant prostate cancer -derived prognosis signature (CRPCPS), for the prediction of recurrence-free survival in prostate cancer. They also validated the signature in eight patient cohorts, comprising more than 1300 patients. Of these six genes, three genes can be used to distinguish localized castration resistant PCa (CRPC) from hormone-sensitive PCa (HSPC). Overall, the manuscript is well written. The interpretation of results and the conclusions are appropriate and work reported in this manuscript is sound and important. It has the potential to direct clinical classification of primary prostate cancer and guide the treatment selection.
A minor concern: this study identified 287 genes that are differentially expressed in CRPC vs HSPC. By applying these 287 genes to TCGA dataset, the team identified 24 genes that are associated with survival and subsequently narrowed the list down to a 6-gene signature that predicts recurrence-free survival. It is concerned that this method may exclude genes that are related to survival but are not among the 287 differentially expressed gene list. Recommend to include this in the discussion.
Author Response
Point 1: This study identified 287 genes that are differentially expressed in CRPC vs HSPC. By applying these 287 genes to TCGA dataset, the team identified 24 genes that are associated with survival and subsequently narrowed the list down to a 6-gene signature that predicts recurrence-free survival. It is concerned that this method may exclude genes that are related to survival but are not among the 287 differentially expressed gene list. Recommend to include this in the discussion. 

Response 1: We agree with the reviewer that “genes that are related to survival but are not among the 287 differentially expressed gene list” are indeed excluded by our analyses. While such genes are important per se, our focus was on those related to castration resistance, a leading cause of prostate cancer mortality. In fact, a total of 1,784 genes were significantly associated with recurrence-free survival (RFS) based on the univariate Cox regression analysis of the training cohort (data not shown). Nevertheless, we have added the following paragraph to the Discussion to address this issue (596-603): “The performance of this 6-gene CRPCPS was improved when compared to previously reported models. For example, the CRPCPS showed a better discrimination capability than three published transcriptomic signatures, consisting of 3, 9, or 22 genes [20,45,46] (Figure S9). We noticed that none of the genes from the three published signatures [20,45,46] was in the 287 CRPC-associated genes identified in our study (Table S2). One of the published signatures was entirely based on genes implicated in the autophagy process [20], whereas the two others were based on genes’ association with the Gleason score and biochemical recurrence-free survival status”.

Reviewer 2 Report
This article of Jun et al. establishes two related molecular scores for prostate cancer: The so-called CRPCPS score based on six genes for the prediction of recurrence-free survival (RFS) and a second one based on three out of these six genes to distinguish between castration-resistant and hormone-sensitive tumors. The article is well written and the authors present extensive analyses, using about ten different cohorts in total for training and validation. This includes Kaplan-Meier analysis, multivariate Cox regression to show that their CRPCPS score is an independent predictor for RFS, receiver operating characteristic (ROC) curves as well as multivariate logistic regression for 3-,5-, and 8-year RFS together with calibration plots.
Despite these efforts, the steps performed for validation are not sufficient and/or not sufficiently discussed. Validation should follow principles as discussed in Royston P and Altman DG, BMC Med Res Methodol 2013;13:33 and the literature cited therein (this also includes literature on validation for logistic regression models).
Major points:
- Strict calibration: For Cox models, e.g., by superimposing predicted and observed survival curves in the validation cohorts. Kaplan-Meier plots as in Fig. 6 provide a rough assessment here. Yet, this is not a strict calibration, could be misleading, and in addition, the curves in Fig. 6 seem to partly differ.
- The authors present calibration plots for logistic regression models for 3-,5-, and 8-year RFS (Fig. 11 c and d). The plots are (only) shown for the training cohort and the combined training and validation cohort (TCGA). The same holds true for the ROC curves in Fig. 4e, 5 c/d, 11b, and S3.
- No rigorous internal validation seems to be performed here, which requires resampling techniques such as bootstrapping or (nested) cross-validation to give a more realistic judgement of model performance on independent samples.
- Since several cohorts in the article are not used in model development, one wonders why these are not used for (external) validation in the above-mentioned analyses (in contrast to the analyses mentioned in point 3 below)?
- One part of external validation of the CRPCPs score - check of discrimination via comparison of hazard ratios – can be performed using Figure 8 and Table 3. These results, including differences between the cohorts and implications, should be discussed in more detail. To better understand differences between the cohorts, Table 1 should at least be extended with data on RFS/follow-up and number of CRPC/HSPC samples. Overall stage, treatment, important inclusion/exclusion criteria and other important variables could also be added here (or in an extra supplemental table).
- As discussed in Royston and Altman (see above), validation analysis for a Cox model could additionally comprise comparison the distribution of the prognostic index (PI) in the different cohorts, regression on the PI in the validation cohorts or calculation of measures of discrimination (e.g., Harrells’s c-index, Royston-Sauerbrei D statistic, Brier score).
The authors should consider performing such kind of analyses. Model performance should be quantified in the abstract and discussion, not just stated. - What was the rationale for splitting the TCGA data at a 7:3-ratio in a training and (internal) validation cohort? Splitting-the-sample approaches generally leads to suboptimal model performance, unless the sample size is very large. This could be improved using resampling methods. In addition, using splines or fractional polynomials in Cox regression could lead to further improvements.
- Separate thresholds for the CRPCPS score were selected for each cohort. Applying the high/low risk groups for clinical practice is therefore infeasible.
Though, categorization leads to a loss of information and moreover, may not be necessary for an application. Are the analyses presented in Table 3 and Fig. 8 based on the categorized or numerical CRPCPS score? - How was RFS defined and did the definitions differ between cohorts? Did the definition of HSPC/CRPC differ between studies?
Minor points:
- For ROC analysis, Gleason score or tumor stage seem to only consider two levels each (≤ 7 vs >7 and T1/2 vs T3/4), while an analysis using all available levels would presumably be preferrable to compare model performance. Still, AUC for the Gleason score is higher than for the CRPCPS score in the internal validation cohort (Fig. S3a)!
Moreover, a more comprehensive comparison of different prediction models could consider the CAPRA score and one or several molecular signatures in references [25-27]. - Why were only GSE70770 and GSE35988 used when developing the diagnostic model? Why was not GSE70769/GSE70770 not used for initial DEG analysis?
- Did the authors perform adjustment for multiple testing (DEG analysis)?
- Line 140: Please clarify how to interpret “…detected with more than one gene…” Is this also related to missing values?
- Line 149: Please replace “preprocessing”. This should not be confused with the preprocessing steps in section 2.2.
- Line 335, Line 416 and Line 445 (Figure S8d): Please state which cohorts were used
- Line 418-421, Line 533-540: Statements are not correct for all cohorts. This and Line 422-434 should be discussed in relation with point 3 (differences between validation cohorts).
- Figure 1:
- TCGA data is already used for univariate Cox regression after DEG analysis (Line 164 -165). This is shown differently in Fig 1.
- “Univariate analysis” should be replaced by “Univariate Cox regression analysis” in the legend.
- LASSO is named twice in the legend
- Figure 2:
- Please clarify the direction of the fold change in a and b
- Legends in b and c are mixed up
- Figure 4 a: How were the horizontal and vertical lines defined?
- Figure 5 a/b: Why do units differ (y-axes)? Adding fold changes, confidence intervals, and samples sizes would allow to better judge the relevance of the differences.
Author Response
Point 1: Strict calibration: For Cox models, e.g., by superimposing predicted and observed survival curves in the validation cohorts. Kaplan-Meier plots as in Fig. 6 provide a rough assessment here. Yet, this is not a strict calibration, could be misleading, and in addition, the curves in Fig. 6 seem to partly differ.
Response 1: We appreciate this comment. In responding to this comment, calibration plots have now been drawn to investigate the model’s performance in all 9 prostate cancer cohorts. We have added the data to Figure 4 and Figure 6 and described them as below (lines 312-314): “The training cohort's calibration plot showed an excellent agreement of 3-, 5-, and 8-year RFS probabilities between the predicted outcomes and actual observations (Figure 4e)”; and in lines 378-380: “The 3-, 5-, and 8-year calibration plots of the internal validation cohort and the entire TCGA cohort resulted in a satisfactory concordance between the predicted and observed RFS probabilities (Figure 6a-b, right panel)”; and in lines 394-396; “The calibration plots from the six external validation cohorts also indicated a good agreement between the estimations and the observations (Figure 6)”.
The curve in Figure 6h slightly differs from those in other cohorts, which has been described as below (lines 393-394): “the prognostic significance of CRPCPS was in the borderline (P = 0.057, Figure 6h)”.
Point 2: The authors present calibration plots for logistic regression models for 3-,5-, and 8-year RFS (Fig. 11 c and d). The plots are (only) shown for the training cohort and the combined training and validation cohort (TCGA). The same holds true for the ROC curves in Fig. 4e, 5 c/d, 11b, and S3.
- No rigorous internal validation seems to be performed here, which requires resampling techniques such as bootstrapping or (nested) cross-validation to give a more realistic judgement of model performance on independent samples.
- Since several cohorts in the article are not used in model development, one wonders why these are not used for (external) validation in the above-mentioned analyses (in contrast to the analyses mentioned in point 3 below)?
Response 2: We appreciate the comment and have now made the following changes in responding to each of these comments:
- With regard to the suggestion that “Add calibration plots of the TCGA training cohort (Fig. 4e), internal validation cohort, and entire TCGA cohort (Fig. S3)”, our response is the same as that to Point 1 described above, with the following changes: 1) For diagnostic model calibration (Fig. 5 c/d), we have added the data to Figure 5 and described them in lines 356-359: “Though the diagnostic model's calibration curves for CRPC prediction in the training and validation cohorts showed some departure between prediction and observation (Figure 5d), the Hosmer-Lemeshow test yielded a nonsignificant statistic (P = 0.955 and 0.295, respectively)”. 2) As for calibration plots adjusted by resampling or bootstrapping, we have added the following sentence to lines 187-190: “Bias-corrected calibration for 3-, 5-, and 8-year RFS probability using 1000 bootstrap resamples was further applied to visualize the consistency between actual and estimated RFS probabilities by the rms (version 6.0.1) R package”.
- A total of four cohorts, including 3 used for DEGs analysis datasets (GSE6811, GSE2443, and GSE28680) and one for diagnostic model validation (GSE35988 ), were excluded for survival analysis due to the lack of survival information for patients. Three of them (GSE6811, GSE2443, and GSE28680) were also excluded for the diagnostic analysis because of their smaller sample sizes (< 50). More details are provided in Table 1 and Table S1.
Point 3: One part of external validation of the CRPCPs score - check of discrimination via comparison of hazard ratios – can be performed using Figure 8 and Table 3. These results, including differences between the cohorts and implications, should be discussed in more detail. To better understand differences between the cohorts, Table 1 should at least be extended with data on RFS/follow-up and number of CRPC/HSPC samples. Overall stage, treatment, important inclusion/exclusion criteria and other important variables could also be added here (or in an extra supplemental table).
Response 3: Following the suggestion to “discuss the hazard ratios”, we have now added the following paragraph to the Discussion section (lines 587-593): “In the multivariate analyses of RFS, the Gleason score and tumor stage showed a greater Hazard Ratio (HR) than the CRPCPS in about half of the cohorts (Figure 8, Table 3), suggesting that the CRPCPS is less effective than the Gleason score and tumor stage in RFS prediction. Nonetheless, the CRPCPS score was significantly correlated with the Gleason score, tumor stages, and lymph node metastasis (Figure 9). In addition, the CRPCPS remained prognostic when patients were stratified by different tumor grade, tumor stage or lymph node status (Figure 10, Figure S6-8)”.
As suggested, more patients’ clinical information, including RFS follow-up time, surgical margins, and treatment, have now been provided (Table 1).
No castration resistance information is available for the patients used for survival analysis (Table 1), except for the Cambridge cohort (GSE70768). For this cohort, the numbers of CRPC and HSPC samples are listed in Table S1 and Figure 5. The inclusion/exclusion criteria of a cohort are provided in Figure 1 and detailed in our response to Point 2.
Point 4: As discussed in Royston and Altman (see above), validation analysis for a Cox model could additionally comprise comparison the distribution of the prognostic index (PI) in the different cohorts, regression on the PI in the validation cohorts or calculation of measures of discrimination (e.g., Harrells’s c-index, Royston-Sauerbrei D statistic, Brier score).
The authors should consider performing such kind of analyses. Model performance should be quantified in the abstract and discussion, not just stated.
Response 4: The literature (Royston P and Altman DG, BMC Med Res Methodol 2013;13:33) has now been cited in the main text.
About the suggestion of “model discrimination analysis in validation cohorts,” we have performed the analyses and added the data as follow:
- In lines 317-319: “The C-index was 0.743 (95% CI: 0.639, 0.847) in the training cohort, and the optimism-corrected C statistic with 1000 bootstrap replications was also 0.743”;
- Lines 380-382: “Additionally, the C-index of the CRPCPS reached 0.70 with 1000 bootstrap iterations (95% CI: 0.56, 0.83) in the internal validation cohort and 0.73 (95% CI: 0.65, 0.82) in the entire TCGA cohort”;
- Lines 396-397: “In addition, the C-index was 0.68 (95% CI: 0.57, 0.78) in the MSKCC cohort and 0.66 (95% CI: 0.57, 0.75) in the Stockholm cohort (GSE70769)”;
- Lines 505-508: “The C-index of our nomogram with 1000 bootstrap resamples reached 0.71 (95% CI: 0.69– 0.88), 0.77 (95% CI: 0.74–0.91), and 0.76 (95% CI: 0.70–0.85) in the training cohort, internal validation cohort, and the entire TCGA cohort, respectively”;
- Abstract, lines 35-37: “Notably, the signature’s robustness was demonstrated by the calibration plot, C-index (0.74), and the 3-, 5- and 8-year area under the receiver operating characteristic curve (0.77, 0.74, and 0.76, respectively, in the training cohort)”;
- Discussion, lines 565-568: “In addition to CRPCPS’s validity in all eight cohorts of more than 1300 patients, the CRPCPS’s C-index was greater than 0.65 in the training cohort, the internal validation cohort, and two other external validation cohorts, indicating excellent discrimination”.
Point 5: What was the rationale for splitting the TCGA data at a 7:3-ratio in a training and (internal) validation cohort? Splitting-the-sample approaches generally leads to suboptimal model performance, unless the sample size is very large. This could be improved using resampling methods. In addition, using splines or fractional polynomials in Cox regression could lead to further improvements.
Response 5: We adopted the 7:3 classification criterion based on previous studies:
- Choi Y et al. Radiomics may increase the prognostic value for survival in glioblastoma patients when combined with conventional clinical and genetic prognostic models. Eur Radiol, 2020.
- Li W et al. Identification and Validation of a Prognostic lncRNA Signature for Hepatocellular Carcinoma. Front Oncol 10:780, 2020.
- Zhou R et al. Immune cell infiltration as a biomarker for the diagnosis and prognosis of stage I-III colon cancer. Cancer Immunol Immunother 68:433-442, 2019.
Although resampling or other approaches (i.e., splines or fractional polynomials) have been applied in model optimization in previous studies, the least absolute shrinkage and selection operator (LASSO) method study is more broadly used for variables selection and optimization. The optimal l, which was estimated from 10-fold cross-validation, resulted in a promising combination, and the optimal combination was finally determined by the Akaike Information Criterion (AIC) using the stepwise method. Details are provided in section 2.4 and section 3.2.
Point 6: Separate thresholds for the CRPCPS score were selected for each cohort. Applying the high/low risk groups for clinical practice is therefore infeasible.
Though, categorization leads to a loss of information and moreover, may not be necessary for an application. Are the analyses presented in Table 3 and Fig. 8 based on the categorized or numerical CRPCPS score?
Response 6: We appreciate the comment. Molecular biomarkers are more variable than clinical parameters such as the Gleason score and tumor stage due to variations generated during sample collection, RNA extraction, and gene expression analysis. Such a variability increases inter-study heterogeneity, as mentioned in lines 668-669: “These heterogeneities made us choose different optimal thresholds for risk stratification.” Therefore, to develop a valid prognostic model using different patient cohorts, we should apply more strict selections and use more patient samples, which we have tried to do. In clinical practice, the threshold separating patients into high- or low-risk groups must be determined by clinical trials. Although the CRPCPS developed in our study showed some unsatisfactory outcomes in a small number of patients, it was generated by a strict selection and validated in a large number of patients from multiple cohorts and thus should be worth further evaluation for clinical practice.
A continuous CRPCPS score was used throughout our study.
Point 7: How was RFS defined and did the definitions differ between cohorts? Did the definition of HSPC/CRPC differ between studies?
Response 7: The recurrence status and time to relapse for each patient were recorded clearly in all the cohorts. Patients without recurrence information or last follow-up time/recurrence time were excluded.
Androgen-independent disease was annotated in only two of the studies analyzed (GSE2443 and GSE6811). Although the CRPC status was not available in other datasets, the samples’ information was easily obtained from their series matrix files. Each sample/tumor assigned as “CRPC/HRPC/AIPC” was included for DEGs identification. As mentioned in the Method section, all the samples used for DEG analysis must be derived from the primary site (i.e., the prostate). For HSPC samples in DEG analysis, we included all those derived from the prostate, regardless of their metastatic statuses. For diagnostic analysis, samples of metastatic CRPC in the validation dataset (GSE35988) were also included due to the rarity of primary CRPC tumors.
Point 8: For ROC analysis, Gleason score or tumor stage seem to only consider two levels each (≤ 7 vs >7 and T1/2 vs T3/4), while an analysis using all available levels would presumably be preferrable to compare model performance. Still, AUC for the Gleason score is higher than for the CRPCPS score in the internal validation cohort (Fig. S3a)!
Moreover, a more comprehensive comparison of different prediction models could consider the CAPRA score and one or several molecular signatures in references [25-27].
Response 8: We appreciate the comment. The two-level Gleason score/tumor stage categorizing (≤ 7 vs >7 and T1/2 vs T3/4) was also used in other parts of our study, and such an approach is in line with published studies. Here are just a few such studies:
- Shi R et al. A novel gene signature-based model predicts biochemical recurrence-free survival in prostate cancer patients after radical prostatectomy. Cancers (Basel) 12, 2019.
- Jiang Y et al. Construction of a set of novel and robust gene expression signatures predicting prostate cancer recurrence. Mol Oncol 12:1559-1578, 2018.
We agree with the reviewer that the Gleason score had higher AUCs than the CRPCPS score in the internal validation cohort (Figure S3). In responding to this comment, we have added the following sentence in lines 372-374: “Although the CRPCPS’ AUCs are smaller than those of the Gleason score, they still displayed good discrimination in the internal validation cohort (all were greater than 0.65, Figure S3b)”.
Regarding the suggestion that “comparison with other signatures,” we have added the data to Figure S9 and Table S5 and described the data different locations: 1) in lines 192-195: “The integrated area under the curves (IAUC) and the hazard ratio (HR) of the model (i.e., CRPCPS) were compared to another three published transcriptomic signatures [20,45,46] by survcomp (version 1.36.1) package. The prediction error was estimated by the pec (version 2020.11.17) R package”. 2) lines 484-492 (section 3.7): “To further evaluate this 6-gene signature's predictive performance, we compared the CRPCPS with three published signatures in the same training cohort. In general, the CRPCPS had an improved discrimination capability. For example, the CRPCPS had higher AUCs at 3-, 5-, and 8-year (0.773, 0.742, and 0.768, respectively) than the signatures from the Wang study [46] (0.638, 0.587, and 0.597, respectively), the Shi study [45] (0.66, 0.577, and 0.623, respectively), and the Hu study [20] (0.697, 0.566, and 0.638, respectively) (Figure S9a-c, Wilcoxon P < 0.001). In addition, the CRPCPS had a smaller predicted error than the three signatures in the predicted error curves (Figure S9d) and a greater hazard ratio than those in the Shi and Hu studies (Table S5)”.
Point 9: Why were only GSE70770 and GSE35988 used when developing the diagnostic model? Why was not GSE70769/GSE70770 not used for initial DEG analysis?
Response 9: Firstly, it needs to be clarified that the Series GSE70770 contains two expression arrays: GSE70768 (CamCap) and GSE70769 (Stockholm). Both GSE70768 and GSE70769 were generated from the same platform, but the GSE70768 contained CRPC samples and was also used for DEG analysis. The GSE0769 cohort was excluded for DEG analysis due to the absence of CRPC samples and the batch effects after combining with the GSE70768 cohort.
Our use of GSE35988 and GSE70770 for developing the diagnostic model is based on the following reasons: (1) Although there was an issue of batch effects after combining GSE70768 and GSE70769, the same platform generated both datasets and thus, the issue of batch effects should be minimal compared to the combination of datasets from different platforms. It should be noted that the number of samples in each combination in our diagnostic analysis must be greater than 50.
Point 10: Did the authors perform adjustment for multiple testing (DEG analysis)?
Response 10: No additional multiple testing was performed in the DEGs analysis for the following reasons:
1) There are differences in DEGs and their fold changes between different platforms (e.g., RNA-Seq vs. microarray, one type of microarray vs. another, etc.). Tumor heterogeneity in CRPCs is more severe than in primary tumors, which also causes variations in DEGs. Therefore, a moderate cut-off value (log2 (fold change) > 0.5), while produces more false positives, was necessary to include more DEGs. The application of an integrated method after the initial DEG identification (see in section 2.3) was necessary to reduce the negative impact of the false positives.
It is not unusual to exclude multiple testing or use less strict cut-off value (i.e., padj < 0.1 or false discovery rate < 0.3) in DEG analysis, and here are just a few examples:
- Sun W et al. Differential Expression Profiles and Functional Prediction of Circular RNAs in Pediatric Dilated Cardiomyopathy. Front Mol Biosci 7: 600170, 2020.
- Stertz L et al. Convergent genomic and pharmacological evidence of PI3K/GSK3 signaling alterations in neurons from schizophrenia patients. Neuropsychopharmacology, 2020.
- Hannen R et al. Comparative transcriptomic analysis of temozolomide resistant primary GBM stem-like cells and recurrent GBM identifies up-regulation of the carbonic anhydrase CA2 gene as resistance factor. Cancers 11, 2019.
- Mock A et al. LOC283731 promoter hypermethylation prognosticates survival after radiochemotherapy in IDH1 wild-type glioblastoma patients. International Journal of Cancer 139: 424-32, 2016.
Point 11: Line 140: Please clarify how to interpret “…detected with more than one gene…” Is this also related to missing values?
Response 11: Each non-specific probe was mapped to more than one gene (not missing value) and thus was excluded in our analysis.
Point 12: Line 149: Please replace “preprocessing”. This should not be confused with the preprocessing steps in section 2.2.
Response 12: We have rephrased the original sentence of “The R package “limma” [36] was applied to preprocess the expression profiles for identifying DEGs between localized CRPC and HSPC tumors” with the following (lines 148-149): “The R package “limma” [36] was applied to identify DEGs between localized CRPC and HSPC tumors”.
Point 13: Line 335, Line 416 and Line 445 (Figure S8d): Please state which cohorts were used.
Response 13: The cohorts’ names have now been added in lines 335 (new line 351), 416 (new line 441), and 445 (new line 472), and Figure S8d.
Point 14: Line 418-421, Line 533-540: Statements are not correct for all cohorts. This and Line 422-434 should be discussed in relation with point 3 (differences between validation cohorts).
Response 14: We have made the following changes in responding to these comments:
Line 418-421: We have rephrased the original sentence of “As expected, the CRPCPS score was positively associated with tumor stage (Figure 9a, Figure S4a-d), Gleason score (Figure 9b, Figure S4e-h), and lymph node status (Figure 9c, Figure S4i-k). However, the CRPCPS score was not significantly associated with patient ages (> vs. ≤ 65 years) in any of the six cohorts (Figure S5)” to the following (lines 443-448): “As expected, a higher CRPCPS score was positively associated with multiple adverse clinicopathological features, including higher tumor stage (Figure 9a, Figure S4a-d), higher Gleason score (Figure 9b, left panel; Figure S4e-f, S4h), and lymph node metastasis (Figure 9c, Figure S4i-j). The exception was patients’age (> vs. ≤ 65 years), which was not significantly associated with the CRPCPS score in any of the six cohorts (Figure S5)”.
Line 533-540: We have rephrased the original sentence of “CRPCPS is an independent prognostic factor for RFS and can predict RFS in PCa patients with a specific clinicopathological parameter. Multivariate analysis demonstrated that the CRPCPS score was an effective risk factor for RFS prediction (Figure 8, Table 3). CRPCPS scores also significantly correlated with tumor stages, lymph node metastasis, and Gleason scores (Figure 9). Analyses with further stratification indicated that CRPCPS remains prognostic in PCa patients with a specific clinicopathological parameter, including tumor stage, lymph node status, and Gleason score (Figure 10, Figure S6-8)” to the following (lines 578-586): “Although unsatisfactory in some analyses in a small number of patients, it is clear that the CRPCPS is an independent prognostic factor for RFS and can predict RFS in PCa patients with a specific clinicopathological parameter. For example, multivariate analysis demonstrated that the CRPCPS score was an effective risk factor for RFS prediction in six of nine PCa cohorts (Figure 8, Table 3). In three of the nine cohorts, the CRPCPS failed to predict RFS. It should be noted, however, the Gleason score, patient’s age at diagnosis, PSA level, and tumor stage also was unable to predict RFS in one of the three cohorts (the Belfast cohort); and one of the six CRPCPS genes (i.e., NPEPL1) was absent in another (the CPC-GENE cohort)”.
Discuss with point 3 (lines 587-595): “In the multivariate analyses of RFS, the Gleason score and tumor stage showed a greater Hazard Ratio (HR) than the CRPCPS in about half of the cohorts (Figure 8, Table 3), suggesting that the CRPCPS is less effective than the Gleason score and tumor stage in RFS prediction. Nonetheless, the CRPCPS score was significantly correlated with the Gleason score, tumor stages, and lymph node metastasis (Figure 9). In addition, the CRPCPS remained prognostic when patients were stratified by different tumor grade, tumor stage or lymph node status (Figure 10, Figure S6-8). Furthermore, an enhanced predictive capability was detected after integrating the CRPCPS with the Gleason score, tumor stage, lymph node status, and age at diagnosis into a nomogram (Figure 11)”.
Point 15: Figure 1:
- TCGA data is already used for univariate Cox regression after DEG analysis (Line 164 -165). This is shown differently in Fig 1.
- “Univariate analysis” should be replaced by “Univariate Cox regression analysis” in the legend.
- LASSO is named twice in the legend.
Response 15: We have made the following changes in responding to these comments:
1) Figure 1 has now been corrected as suggested.
2) The original phrase of “Univariate analysis” has now been replaced by “Univariate Cox regression analysis” (line 133).
- We apologize for this error in the figure legend. The repeated word “LASSO” has now been deleted.
Point 16: Figure 2:
- Please clarify the direction of the fold change in a and b.
- Legends in b and c are mixed up.
Response 16: We have made the following changes in responding to these comments:
1) The direction of the fold change has now been added to Figure 2a and described in lines 276-277 as the following: “The gradual color ranging from blue to red represents the changing process from down- to up-regulation (CRPC vs. HSPC)” for Figure 2b.
- We apologize for the mix-up in figure legends. We have now fixed them.
Point 17: Figure 4 a: How were the horizontal and vertical lines defined?
Response 17: The original horizontal lines were defined as the median CRPCPS score, and the vertical lines representing the borderline dividing patients into high- and low-risk groups. The Y-axe value towards the vertical lines intersected with the curve was determined by the function of “surv_cutpoint” (described in section 2.4). To avoid misunderstanding the grouping, the original horizontal lines in Figure 4a have now been deleted.
Point 18: Figure 5 a/b: Why do units differ (y-axes)? Adding fold changes, confidence intervals, and samples sizes would allow to better judge the relevance of the differences.
Response 18: The y-axes represent the mRNA expression levels. The differences in mRNA expression value between the training cohort and the validation cohorts are mainly caused by the platform's variations (i.e., one type of microarray vs. another in this case) and the normalization methods.
Regarding the suggestion to “add fold changes for these six genes”, the fold changes of these six genes were included in Table S2. We have now added the number of samples in Figure 5a/b, and the confidence intervals of AUC in Figure 5 c/d.

Reviewer 3 Report
Summary
The manuscript described a bioinformatics workflow to extensively study the differential expressional level of the genes and the differential expression profile of gene panels, to identify the predictive molecular signatures for predicting RFS and CRPC. The authors also assess the performance of the identified six- and three-gene panel-based gene expression signatures in six independent patient cohorts and the robustness of the identified molecular signatures was validated by the independent data sets. in assisting the manual curation of annotations for prokaryotic genomes. The study reported by authors provided a new potential molecular prognostic tool, six-gene and three-gene panel, to predict and assess RFS in PCa pa- 611 patients. The gene-panel-based gene expression profile signature or biomarker discovered by the study could be used to classify or group patients into different sub-groups for better personal precision treatment and diagnosis.
The authors did hard work in downloading, processing and analyzing a large volume of data sets in their study. I have no major comments but some minor comments and questions for the authors.
Minor comments:
- Line154 in section 2.3. Identification of CRPC-associated genes, authors need to explain why the fold change |log2| > 0.5 is used as the cutoff to select more robust differentially expression genes? Why not to use more strict fold change cutoff to select more robust DEGs with potential smaller false positive rate, e.g. |log2| > 1, |log2| > 2?
- In section 2. Materials and Methods, the authors must provide the version numbers of all R packages and other software tools and databases used in the study. Given the same input data, the different versions of the same package/software/database may output the different results.
- Line14 in section ‘Simple Summary’, the full name of CPRC is not clear.
- Line17 in section ‘Simple Summary’, the full name of abbreviation PCa is not clear.
Author Response
Point 1: Line154 in section 2.3. Identification of CRPC-associated genes, authors need to explain why the fold change |log2| > 0.5 is used as the cutoff to select more robust differentially expression genes? Why not to use more strict fold change cutoff to select more robust DEGs with potential smaller false positive rate, e.g. |log2| > 1, |log2| > 2?
Response 1: The use of fold change |log2| > 0.5 rather than more strict values (i.e., |log2| > 1 or |log2| > 2) for DEGs analysis is based on the following reasons:
1) Compared to the RNA-Seq platform, transcriptomic data from the microarray platform are more variable among different studies. Taken together with the high heterogeneity of CRPC tumors, we used a moderate cut-off value (log2 (fold change) > 0.5) to select genes, which would prevent excluding genes that were important but did not show consistent higher-fold differences among different studies. Our use of this approach could have led to more genes than necessary. However, we then used an integrated method to reduce the negative impact of potential false positives. Although the fold change of 254 CRPC-associated genes (Table S2) after integration was lower (< 2), many showed higher fold changes (≥ 2) in at least two datasets (e.g., COL5A2, THBS2, CDC20) (Figure 2b, Table S2).
2) It is not unusual to use log2 (fold change) < 1 in the field. Here are a few examples:
- Köster T et al. Regulation of pri-miRNA processing by the hnRNP-like protein AtGRP7 in Arabidopsis. Nucleic Acids Res 42:9925-9936, 2014.
- Mock A et al. LOC283731 promoter hypermethylation prognosticates survival after radiochemotherapy in IDH1 wild-type glioblastoma patients. Int J Cancer 139:424-432, 2016.
- Hannen R et al. Comparative transcriptomic analysis of temozolomide resistant primary GBM stem-like cells and recurrent GBM identifies up-regulation of the carbonic anhydrase CA2 gene as resistance factor. Cancers (Basel) 11, 2019.
- Cojocneanu R et al. Plasma and tissue specific miRNA expression pattern and functional analysis associated to colorectal cancer patients. Cancers (Basel) 12, 2020.
Point 2: In section 2. Materials and Methods, the authors must provide the version numbers of all R packages and other software tools and databases used in the study. Given the same input data, the different versions of the same package/software/database may output the different results.
Response 2: We appreciate the reviewer for pointing this out. The version numbers of different R packages used in our study have now been provided.
Point 3: Line14 in section ‘Simple Summary,’ the full name of CPRC is not clear.
Response 3: The abbreviation of “CPRC” has been replaced by “Castration-resistant PCa” in the section of ‘Simple Summary.’
Point 4: Line17 in section ‘Simple Summary,’ the full name of abbreviation PCa is not clear.
Response 4: The abbreviation “PCa” has been replaced by “Prostate cancer (PCa)” as suggested.

Round 2
Reviewer 2 Report
Appreciating the various improvements and detailed response to the comments by the authors, there are still several point to consider:
- Lines 356-359/Figure 5c:
- The Hosmer-Lemeshow test should not be used, due to its low power and arbitrary binning. See, e.g., Hosmer, D. W.; Hosmer, T.; le Cessie, S. & Lemeshow, S. A comparison of goodness-of-fit tests for the logistic regression model. Statistics in Medicine 16, 965-980 (1997). This paper also introduces an improved approach and there are also various other alternatives.
- The clear difference between ideal and apparent/bias-corrected curves in Figure 5c raises the question for some sort of recalibration of the model, see e.g., the literature cited in Royston, P., Altman, D.G. External validation of a Cox prognostic model: principles and methods. BMC Med Res Methodol 13, 33 (2013).
- Legend in both plots of Figure 5c: “Ideal” and “Apparent” are mixed up. Legend below the figure is correct.
- Measures of discrimination should not only be presented for some, but all of the validation cohorts, especially for the Cox model with CRPCs score only (lines 396-397 and 565-568). Some sort of summary of these measures (e.g., range or minimum) over all validation cohorts should also be included in the abstract.
Besides, even if a c-index of 0.65 or even less may be acceptable for this kind of application, it is far from being optimal and the wording should be downgraded accordingly. - The check of discrimination via comparison of hazard ratios (HRs) should be adapted, considering the following points:
- Lines 587-593: Comparing HRs between variables with different units/scales is meaningless, and one cannot draw conclusions about the relative effectiveness of the different variables using such an approach.
- Rather, one part of the validation can be performed by comparing the HRs of the same variable (in particular the CRPCPS score) between training and validation cohorts.
- In univariate analysis, HRs of the CRPCPS score may or be acceptably close (Figure 8); this could be further supported by univariate analyses in the other validation cohorts. However, this may require some normalization or standardization approach to obtain the same scale for the CRPCPS score in the different cohorts, given the different platforms.
- In multivariate analysis, HRs of the CRPCPS score in multivariate analysis clearly differ between training and validation cohort (Figure 8), arguing against a successful validation of the multivariate model.
The HRs for the Gleason score also strongly disagree, with huge confidence interval in the validation cohort. One reason for this could be the use of the suboptimal splitting-the-sample approach.
- The results in Figure 8 and Table 3 seem to contradict the conclusion that CRPCPS score is clearly an independent prognostic factor for RFS (e.g., lines 533-540, lines 578-586). E.g., the CRPCPS score is not significant in 50% of the validation cohorts, for which the Gleason score was used as a covariate and the two largest validation cohorts (internal validation cohort, Belfast) show insignificant results.
The CRPCPS score is indeed significant in multivariate analysis of the MSKCC and the Stockholm cohort. However, although listed in Table 1, Gleason score was not used as a covariate here (why?)
Overall, downgrading of wording seems to be necessary in this context (e.g., lines 533-540, lines 578-586). - Lines 484-492/Figure S9: The different signatures were only compared in the training cohort and no correction for overfitting seems to be considered here. Therefore, a higher AUC for the CRPCPs is not surprising. A fair comparison between the methods should be ensured. Values in lines 37 in the abstract should be adapted accordingly.
- Table S5: Unless the different signatures use the same scale, the hazard ratios cannot be compared directly. Moreover, the analysis was again only performed in the training data. Since the focus of this article is not the validation of the other signatures, my suggestion would be to omit this table, unless a reliable conclusion can be drawn.
- Differences in HSPC/CRPC definition between cohorts should be specified in the text (see Response 7).
- Please add the number/percentage of RFS events to Table 1.
Author Response
Point 1: Lines 356-359/Figure 5c:
- The Hosmer-Lemeshow test should not be used, due to its low power and arbitrary binning. See, e.g., Hosmer, D. W.; Hosmer, T.; le Cessie, S. & Lemeshow, S. A comparison of goodness-of-fit tests for the logistic regression model. Statistics in Medicine 16, 965-980 (1997). This paper also introduces an improved approach and there are also various other alternatives.
- The clear difference between ideal and apparent/bias-corrected curves in Figure 5c raises the question for some sort of recalibration of the model, see e.g., the literature cited in Royston, P., Altman, D.G. External validation of a Cox prognostic model: principles and methods. BMC Med Res Methodol 13, 33 (2013).
- Legend in both plots of Figure 5c: “Ideal” and “Apparent” are mixed up. Legend below the figure is correct.
Response 1: We appreciate the reviewer’s expertise in the methodology, although the use of the Hosmer-Lemeshow test for calibration examination is relatively common in the literature. Here are some recent publications in this regard:
- Cao Y et al. Development and validation of MMR prediction model based on simplified clinicopathological features and serum tumour markers. EBioMedicine 61, 2020.
- Takura T et al. Development of a predictive model for integrated medical and long-term care resource consumption based on health behaviour: application of healthcare big data of patients with circulatory diseases. BMC Med 19, 2021.
- Martinez-Zayas G et al. A prediction model to help with oncologic mediastinal evaluation for radiation: HOMER. Am J Respir Crit Care Med 201:212-223, 2020.
- Liu C et al. A nomogram for predicting mortality in patients with COVID-19 and solid tumors: a multicenter retrospective cohort study. J Immunother Cancer 8, 2020.
In addressing this comment, we have now re-assessed the diagnostic model calibration using another approach, i.e., the Brier score, which has been described in the publications below:
- Harrell F et al. Regression modeling strategies: with applications to linear models, logistic and ordinal regression, and survival analysis, 2nd ed. Springer International Publishing: New York, 2015.
- Gude-Sampedro F et al. Development and validation of a prognostic model based on comorbidities to predict Covid-19 severity. A population-based study. Int J Epidemiol. 2020.
Calibration was also assessed using the Brier score. Accordingly, we have added the following sentence to line 373-374 on page 12: “… … and the CRPCPS showed excellent Brier scores (0.03 and 0.09 for the training and validation cohorts respectively)”.
Accordingly, we have added the related methods to the Methods section as below (line 222-223 on Page 6): “Calibration was examined by using a Hosmer-Lemeshow test [49] and the Brier score [50]”.
For both plots of Figure 5c, “Ideal” and “Apparent” have now been corrected as suggested.
Point 2: Measures of discrimination should not only be presented for some, but all of the validation cohorts, especially for the Cox model with CRPCs score only (lines 396-397 and 565-568). Some sort of summary of these measures (e.g., range or minimum) over all validation cohorts should also be included in the abstract.
Besides, even if a c-index of 0.65 or even less may be acceptable for this kind of application, it is far from being optimal and the wording should be downgraded accordingly.
Response 2: As suggested by the reviewer, we have performed the analyses and rephrased some wording. The following changes have been made in different parts of the manuscript:
Lines 396-397: the original sentence of “In addition, the C-index was 0.68 (95% CI: 0.57, 0.78) in the MSKCC cohort and 0.66 (95% CI: 0.57, 0.75) in the Stockholm cohort (GSE70769)” has been changed the following (line 411-415 on Page 14): “In addition, the C-index was 0.68 (95% CI: 0.57, 0.78) in the MSKCC cohort, 0.66 (95% CI: 0.57, 0.75) in the Stockholm cohort (GSE70769), and 0.62 (95% CI: 0.49, 0.75) in the Cambridge cohort. Less ideal C-indices were observed in the Belfast cohort (C-index: 0.59; 95% CI, 0.52-0.67), the GSE54460 cohort (C-index: 0.58; 95% CI, 0.50-0.67), and the CPC_GENE cohort (C-index: 0.55; 95% CI, 0.44-0.74)”.
Lines 565-568: The original sentence of “Our 6-gene CRPCPS model showed excellent calibration and discrimination. In addition to CRPCPS’s validity in all eight cohorts of more than 1300 patients, the CRPCPS’s C-index was greater than 0.65 in the training cohort, the internal validation cohort, and two other external validation cohorts, indicating excellent discrimination” has been changed to the following (line 588-591 on Page 23): “Our 6-gene CRPCPS model showed a reasonable performance in both calibration and discrimination. In addition to CRPCPS’s validity in all eight cohorts of more than 1300 patients, the C-index was acceptable in half of the eight cohorts, suggesting an acceptable prediction capability”.
Line 35-37 in the Abstract: The original sentence of “the signature’s robustness was demonstrated by the calibration plot, C-index (0.74), and the 3-, 5- and 8-year area under the receiver operating characteristic curve (0.77, 0.74, and 0.76, respectively, in the training cohort)” has been changed to the following (lines 37-39): “The signature’s robustness was demonstrated by the C-index (0.55-0.74) and the calibration plot in all nine cohorts and the 3-, 5- and 8-year area under the receiver operating characteristic curve (0.67-0.77) in three cohorts”.
Point 3: The check of discrimination via comparison of hazard ratios (HRs) should be adapted, considering the following points:
- Lines 587-593: Comparing HRs between variables with different units/scales is meaningless, and one cannot draw conclusions about the relative effectiveness of the different variables using such an approach.
- Rather, one part of the validation can be performed by comparing the HRs of the same variable (in particular the CRPCPS score) between training and validation cohorts.
â… . In univariate analysis, HRs of the CRPCPS score may or be acceptably close (Figure 8); this could be further supported by univariate analyses in the other validation cohorts. However, this may require some normalization or standardization approach to obtain the same scale for the CRPCPS score in the different cohorts, given the different platforms.
â…¡. In multivariate analysis, HRs of the CRPCPS score in multivariate analysis clearly differ between training and validation cohort (Figure 8), arguing against a successful validation of the multivariate model.
The HRs for the Gleason score also strongly disagree, with huge confidence interval in the validation cohort. One reason for this could be the use of the suboptimal splitting-the-sample approach.
Response 3: We have now made the following changes in responding to each of these comments:
For comments “a” and “b,” in Lines 587-593: We have rephrased the original sentence of “the Gleason score and tumor stage showed a greater Hazard Ratio (HR) than the CRPCPS in about half of the cohorts (Figure 8, Table 3), suggesting that the CRPCPS is less effective than the Gleason score and tumor stage in RFS prediction. Nonetheless, the CRPCPS score was significantly correlated with the Gleason score, tumor stages, and lymph node metastasis (Figure 9). In addition, the CRPCPS remained prognostic when patients were stratified by different tumor grade, tumor stage or lymph node status (Figure 10, Figure S6-8)” to the following (lines 606-615): “the CRPCPS’ hazard ratios (HRs) were 2.72 (95% CI: 1.66, 4.48) in the training cohort, and 2.03 (95% CI: 1.41, 2.92) in the entire TCGA cohort (Figure 8). Similar HRs were found in three of the seven additional validation cohorts, including the MSKCC cohort (HR = 2.15; 95% CI: 1.30-3.56), the Stockholm cohort (HR = 2.68; 95% CI: 1.34-5.37), and the GSE54460 cohort (HR = 1.74; 95% CI: 1.07-2.84)). These findings suggest that the CRPCPS is capable of RFS prediction in PCa. The CRPCPS score was significantly correlated with the Gleason score, tumor stages, and lymph node metastasis (Figure 9). In addition, the CRPCPS remained prognostic when patients were stratified by different tumor grade, tumor stage or lymph node status (Figure 10, Figure S6-8)”.
For comment “I”, we actually did the univariate analysis in each of the 9 cohorts. In addition to the HRs for the three cohorts presented in Figure 8 (2.18-2.72), HRs for the remaining 6 cohorts were similar, ranging from 1.66 to 3.08. Furthermore, because the CRPCPS score is a continuous variable, the HRs of the CRPCPS scores among different cohorts would not be influenced by the grouping method. Therefore, we think that the HRs of CRPCPS scores should be comparable among different cohorts. We have added the following sentence to lines 438-440 on page 16: “… … with hazard ratios (HRs) of 2.72, 2.18 and 2.51 respectively. The HRs of the CRPCPS scores for the remaining 6 cohorts ranged from 1.66 to 3.08 (P < 0.05 in all but one cohort)”.
For comment “II”, we’d like to emphasize that in multivariate analysis, even well-established prognostic markers such as the Gleason score and tumor stage cannot significantly predict survival in every cohort. Many factors influence the outcome, including the number of samples and potential interaction between different factors. In the current study, although there was a departure in CRPCPS score’s HRs between the training cohort and the internal validation cohort, the prediction power of CRPCPS was detected in five of the nine cohorts, which is much better than expected.
Finally, although we agree with the reviewer that a smaller confidence interval could be achieved by re-grouping samples for the Gleason score in the internal validation cohort (Figure 8), our current splitting of Gleason scores >7 vs. < or = 7 is based on their association with patient RFS. It is well known that patients with >7 Gleason scores have significantly worse RFS than those with 7 or below.
Point 4: The results in Figure 8 and Table 3 seem to contradict the conclusion that CRPCPS score is clearly an independent prognostic factor for RFS (e.g., lines 533-540, lines 578-586). E.g., the CRPCPS score is not significant in 50% of the validation cohorts, for which the Gleason score was used as a covariate and the two largest validation cohorts (internal validation cohort, Belfast) show insignificant results.
The CRPCPS score is indeed significant in multivariate analysis of the MSKCC and the Stockholm cohort. However, although listed in Table 1, Gleason score was not used as a covariate here (why?)
Overall, downgrading of wording seems to be necessary in this context (e.g., lines 533-540, lines 578-586).
Response 4: The number of patients was often small at such a level of categorical variables (Gleason score, age at diagnosis, PSA level, tumor stage, lymph node status, and the surgical margins). For example, for the MSKCC and the Stockholm cohorts, which were mentioned by the reviewer, only 13 and 15 patients, respectively, had Gleason scores of >7. The Gleason score was therefore excluded for further analysis in these two cohorts. Some variables had even smaller sample sizes. For example, only six patients had lymph node metastases in the Cambridge cohort. When all the seven variables, when available, were included in the multivariate analysis, the CRPCPS was significant in three of the nine cohorts (i.e., the training cohort, the entire TCGA cohort, and the GSE54460 cohort).
In better addressing this comment, we re-set the minimum sample size for exclusion to 20 for all variables; and re-did the multivariate analysis. Now the CRPCPS is significant in 5 of the 9 cohorts (compared to 6 of 9 in the previous version). We have now added the following sentences to the following (lines 444-445): “In all the nine cohorts, the CRPCPS was significant in 5 of them when variables with a sample size smaller than 20 were excluded for analysis”; lines (447-450): ”When all the seven variables, when available, were included in the multivariate analysis, the CRPCPS was significant in three of the nine cohorts (i.e., the training cohort, the entire TCGA cohort, and the GSE54460 cohort)”.
Accordingly, we have now replaced Table 3 with the new data. We have also revised the sentence in lines 578-586 to the following (lines 601-605), including downgrading words: “Although unsatisfactory in some analyses, the CRPCPS appears to be an independent prognostic factor for RFS and can predict RFS in PCa patients with a specific clinicopathological parameter. For example, multivariate analysis demonstrated that the CRPCPS score was an effective risk factor for RFS prediction in five of nine PCa cohorts (Figure 8, Table 3)”.
Point 5: Lines 484-492/Figure S9: The different signatures were only compared in the training cohort and no correction for overfitting seems to be considered here. Therefore, a higher AUC for the CRPCPs is not surprising. A fair comparison between the methods should be ensured. Values in lines 37 in the abstract should be adapted accordingly.
Response 5: In responding to this comment, we have used the entire TCGA cohort and the MSKCC cohort to compare the CRPCPS with three published signatures. Use of these two cohorts is based on the following reasons: 1) They have the largest numbers of patients (n = 436 in the entire TCGA cohort and n = 140 in the MSKCC cohort); 2) An external validation cohort (i.e., the MSKCC cohort) should be included; and 3) Each of the cohorts must have been used in at least two studies. The entire TCGA cohort was used to develop or test all the three published signatures, and the MSKCC cohort was used in the Wang study and the Shi study. In addition, the same formula was used in these two cohorts to obtain the risk scores, which makes the comparison more objective and reliable (see Response 6 below). Accordingly, we have added the data to Figure S9 and made the following revisions:
We have revised the original sentences in lines 484-492 to the following (lines 506-516): “... … we compared the CRPCPS with three published signatures in the entire TCGA cohort (n = 436) and the MSKCC cohort (n = 140), two large cohorts that had been used in at least two of the three published studies. In the entire TCGA cohort, the CRPCPS’ AUCs at 3-, 5-, and 8-year were 0.77, 0.74, and 0.73, respectively, which were greater than those in the Wang study [46] (0.64, 0.59, and 0.60, respectively), the Shi study [45] (0.66, 0.58, and 0.62, respectively), and the Hu study [20] (0.70, 0.57, and 0.64, respectively) (Figure S9a-c, left panel, Wilcoxon P < 0.001). In addition, the CRPCPS had a smaller predicted error than the three signatures in the predicted error curves (Figure S9d, left panel). In the MSKCC cohort, while the Shi signature performed better than the CRPCPS, the CRPCPS performed better than those of the Wang and Hu signatures (Figure S9a-d, right panel)”.
The sentence in line 37 in the abstract has been revised accordingly (see Response 2).
Point 6: Table S5: Unless the different signatures use the same scale, the hazard ratios cannot be compared directly. Moreover, the analysis was again only performed in the training data. Since the focus of this article is not the validation of the other signatures, my suggestion would be to omit this table, unless a reliable conclusion can be drawn.
Response 6: The CRPCPS indeed used the same scale as the other signatures because the same formula was used to calculate the risk scores for all the four signatures in the same cohorts. Therefore, the hazard ratios of these four signatures can be compared directly.
As suggested, we have now deleted Table S5.
Point 7: Differences in HSPC/CRPC definition between cohorts should be specified in the text (see Response 7).
Response 7: Following the suggestion, we have now added the following sentences to the Methods section (lines 110-117 on page 3): “In general, a CRPC is defined by the consecutive elevation of serum PSA levels and/or the appearance of progressive measurable diseases such as enlargement of a primary tumor or the detection of metastasis. Of the 5 datasets (including the GSE35988 dataset below) with CRPC tumors, only two provided CRPC definition (GSE2443 and GSE6811), and both datasets used one or more of the above criteria. In the other three datasets, a patient’s CRPC status was provided without specifying how it was defined. CRPC and HSPC statuses of all patients were retrieved from the series matrix files of all datasets.”
Point 8: Please add the number/percentage of RFS events to Table 1.
Response 8: As suggested, the number/percentage of RFS events have now been added to Table 1.
We would like to mention that, although the TCGA training cohort had a lower percentage of recurrence, it was still used for developing the prognostic model because of its larger sample size and the availability of extensive patients’ information.

Round 3
Reviewer 2 Report
I agree with the bulk of changes and comments by the authors, and would suggest few additional modifications:
1. Lines 369-374: A low Brier score may not imply high calibration, since it incorporates components of discrimination and calibration. My suggestion would be to add p-values of Spiegelhalter’s z test, which is related to the calibration component of the Brier score, and in turn omit the Hosmer-Lemeshow test, given its low power and arbitrary binning.
2. Abstract, line 435, and lines 601-605: With the CRPCS score being only significant in about half of the cohorts, it is at least questionable whether the score can really be considered as an independent risk factor. Please omit or further downgrade this strong statement, such that it better reflects the situation in the validation cohorts.
3. Line 595: Please replace “excellent” by “acceptable” or similar (cf lines 590-591).
4. Lines 618-619: Please downgrade wording, since the Shi signature performed better than the CRPCPS in the validation cohort (MSKCC).
Again, a better performance of CRPCPS compared to the other published scores on the entire TCGA cohort is not surprising, since 70% of the TCGA samples were used for training of the CRPCPS. Performance measures computed on the entire TCGA (or TCGA training cohort) are therefore irrelevant for the comparison of the different signatures.
Author Response
Point 1: Lines 369-374: A low Brier score may not imply high calibration, since it incorporates components of discrimination and calibration. My suggestion would be to add p-values of Spiegelhalter’s z test, which is related to the calibration component of the Brier score, and in turn omit the Hosmer-Lemeshow test, given its low power and arbitrary binning.
Response 1: As suggested by the reviewer, we have now re-assessed the diagnostic model’s calibration using the Spiegelhalter’s z test, replacing the previously used Hosmer-Lemeshow test. The following changes have been made in different parts of the manuscript:
Lines 369-374: The original sentence of “Although the diagnostic model's calibration curves for CRPC prediction in the training and validation cohorts displayed some departure between prediction and observation (Figure 5d), the Hosmer-Lemeshow test yielded a nonsignificant statistic (P = 0.955 and 0.295 for the training and validation cohorts respectively), and the CRPCPS showed excellent Brier scores (0.03 and 0.09 for the training and validation cohorts, respectively)” has been changed to the following (lines 369-374 on Page 12): “Although the diagnostic model's calibration curves for CRPC prediction in the training and validation cohorts displayed some departure between prediction and observation (Figure 5d), the CRPCPS showed low Brier scores (0.03 and 0.09 for the training and validation cohorts, respectively), and the Spiegelhalter z test yielded a nonsignificant statistic (P = 0.86 and 0.76 for the training and validation cohorts respectively)”.
Lines 222-223: The original sentence of “Calibration was examined by using a Hosmer-Lemeshow test [49] and the Brier score [50]” has been changed to the following (lines 222-223 on Page 6): “Calibration was examined by using the Brier score [49] and the Spiegelhalter z test [50]”.
Point 2: Abstract, line 435, and lines 601-605: With the CRPCS score being only significant in about half of the cohorts, it is at least questionable whether the score can really be considered as an independent risk factor. Please omit or further downgrade this strong statement, such that it better reflects the situation in the validation cohorts.
Response 2: We have now made the following changes in responding to these comments:
Lines 21 in the Simple Summary: The original sentence of “The CRPCPS was an independent predictor of RFS … …” has been changed to the following (line 21): “The CRPCPS predicted RFS in 5/9 cohorts in the multivariate analysis … …”.
Line 35 in the Abstract: The original sentence of “The CRPCPS was an independent predictor of RFS … …” has been changed to the following (line 35): “The CRPCPS predicted RFS in 5/9 cohorts in the multivariate analysis … …”.
Line 435 in the section 3.5: The original sentence of “The CRPCPS is an independent risk factor for RFS in PCa” has been changed to the following (line 435): “The CRPCPS is prognostic of RFS in PCa”.
Lines 601-605: The original sentence of “Although unsatisfactory in some analyses, the CRPCPS appears to be an independent prognostic factor for RFS and can predict RFS in PCa patients with a specific clinicopathological parameter. For example, multivariate analysis demonstrated that the CRPCPS score was an effective risk factor for RFS prediction in five of nine PCa cohorts (Figure 8, Table 3)” has been changed to the following (lines 601 to 605): “The CRPCPS is prognostic of RFS in PCa patients and can predict RFS in PCa patients with a specific clinicopathological parameter. For example, multivariate analysis demonstrated that the CRPCPS score was an effective risk factor for RFS prediction independent of multiple known prognostic factors in five of nine PCa cohorts (Figure 8, Table 3)”.
Point 3: Line 595: Please replace “excellent” by “acceptable” or similar (cf lines 590-591).
Response 3: Changed as suggested.
Point 4: Lines 618-619: Please downgrade wording, since the Shi signature performed better than the CRPCPS in the validation cohort (MSKCC).
Again, a better performance of CRPCPS compared to the other published scores on the entire TCGA cohort is not surprising, since 70% of the TCGA samples were used for training of the CRPCPS. Performance measures computed on the entire TCGA (or TCGA training cohort) are therefore irrelevant for the comparison of the different signatures.
Response 4: We have now revised the sentence in lines 618-619 to the following (lines 617-619), including downgrading words: “The performance of this 6-gene CRPCPS was improved when compared to some previously reported models. For example, the CRPCPS showed a more promising discrimination capability than two of the three published transcriptomic signatures … …”.
Round 4
Reviewer 2 Report
I agree with the authors' changes and suggest to publish the paper in its present form.